

# Analysis of atmospheric CH$_4$ in Canadian Arctic and estimation of the regional CH$_4$ fluxes

Misa Ishizawa[1], Douglas Chan[1], Doug Worthy[1], Elton Chan[1], Felix Vogel[1] and Shamil Maksyutov[2]

[1]Environment and Climate Change Canada, Toronto, M3H 5T4, Canada
[2]National Institute for Environmental Studies, Tsukuba, 305-8506, Japan

*Correspondence to*: Misa Ishizawa (misa.ishizawa@canada.ca)

**Abstract.** The Canadian Arctic has the potential for enhanced atmospheric methane (CH$_4$) source regions as a response to the ongoing global warming. Current bottom-up and top-down estimates of the regional CH$_4$ flux range widely. This study analyses the recent observations of atmospheric CH$_4$ from five arctic monitoring sites and presents estimates of the regional CH$_4$ fluxes for 2012−2015. The observational data reveal sizeable synoptic summertime enhancements in the atmospheric CH$_4$ that are clearly distinguishable from background variations, which indicate strong regional fluxes (mainly wetland and biomass burning CH$_4$ emissions) around Behchoko and Inuvik in the western Canadian Arctic. Multiple regional Bayesian inversion modelling systems are applied to estimate fluxes for the entire Canadian Arctic and show relatively robust results in amplitude and temporal variations even across different transport models, prior fluxes and sub-region masking. The estimated mean total CH$_4$ annual flux for the Canadian Arctic is 1.8±0.6 TgCH$_4$ yr$^{-1}$. The flux estimate in this study is partitioned into biomass burning, $0.3 \pm 0.1$ TgCH$_4$ yr$^{-1}$, and the remaining natural (wetland) flux $1.5 \pm 0.5$ TgCH$_4$ yr$^{-1}$. The estimated summertime natural CH$_4$ fluxes show clear inter-annual variability that is positively correlated with surface temperature anomalies. This indicates that the hot summer weather conditions stimulate the wetland CH$_4$ emissions. More data and analysis are required to statistically characterise the dependence of regional CH$_4$ fluxes on climate in the Arctic. These Arctic measurement sites should help quantify the inter-annual variations and long-term trends in CH$_4$ emissions in the Canadian Arctic.

## 1 Introduction

Atmospheric Methane (CH$_4$) is one of the principal greenhouse gases with a global warming potential (GWP), 34 times stronger then CO$_2$ over a time period of 100 years, and 96 times over 20 years (Gasser et al., 2017). The atmospheric CH$_4$ level has increased to twice the level of the pre-industrial era, about 722 ppb to 1803 ppb in 2011 (Ciais et al., 2013). The Arctic natural/wetland CH$_4$ emission is an area of interest as it is a potentially growing CH$_4$ source under climate change (AMAP, 2015). The Arctic is mainly continuous permafrost that contains large quantities of soil carbon, ~1700 PgC, (Tarnocai et al., 2009), which is highly vulnerable under the globally warming climate. However, there is only a low confidence in the exact magnitude of CO$_2$ and CH$_4$ emissions caused by the carbon lost and whether the thawing carbon will decompose aerobically to release CO$_2$ or anaerobically to release



$CH_4$ (Ciais et al. 2013). Overall, the natural $CH_4$ flux estimates remain largely uncertain in higher northern latitudes (Kirschke et al., 2013;Saunois et al., 2016).

Bottom-up estimates from wetland methane models in WETCHIMP, some of which are also used for fundamental climate change research within the Coupled Model Intercomparison Phase 6 (CMIP6), show large

discrepancies in the spatial distribution of wetlands, as well as the magnitude (Melton et al., 2013). The wetland models define the extent of existing ecosystems and wetland extents from ground-based inventory and/or space-based information. In the higher latitudes, the limited ground-based information has hindered the mapping of wetland. Recently, remote sensing has been providing more information, but the high-latitude wetland extent still has large uncertainties (Olefeldt et al., 2016;Thornton et al., 2016).

In addition to uncertainty in wetland extent, other factors affecting high-latitude wetland emissions in different models still remain. A recent inter-comparison of $CH_4$ wetland models (Poulter et al., 2017) in which all models used the same wetland extent, Surface Water Microwave Product Series (SWAMPS) (Schroeder et al, 2015) with Global Lakes and Wetland Database (GLWD) (Lehner and Döll, 2004) and same meteorological data (CRU-NCEP v4.0 reconstructed climate data) to drive their models showed a range in estimated $CH_4$ emission for North

American Boreal/Arctic region which remains larger than that for other regions in the world. This large range of the $CH_4$ emissions for North American Boreal/Arctic region indicates the uncertainty in our current understanding of physical and biogeochemical processes that contribute to wetland $CH_4$ emissions.

There have been many studies on $CH_4$ emission using bottom-up and top-down methods and Saunois et al. (2016) provide a thorough review of the different studies. In general, the bottom-up flux estimates for the northern

high latitudes from biogeochemical $CH_4$ models have large variations, and the mean estimate is much higher than the top-down estimates from the inverse modelling (Saunois et al., 2016). For the Boreal North America region including Alaska and the Hudson Bay Lowlands (HBL, the second largest boreal wetland in the world), the bottom-up mean estimate is ~32 $TgCH_4$ $yr^{-1}$, with a wide range from 15 to 60 $TgCH_4$ $yr^{-1}$. On the other hand, the top-down estimate is ~12 $TgCH_4$ $yr^{-1}$ with a narrower range from ~7 to 21 $TgCH_4$ $yr^{-1}$.

Top-down atmospheric inverse modelling infers the fluxes with observed atmospheric concentrations as constraint using different optimizations, including Bayesian (e.g. Thompson et al., 2017; Lin et al., 2003), 4D variational optimization (4Dvar) (e.g. Bergamaschi et al., 2013; Bousquet et al., 2011), ensemble Kalman filter (EnKF) (e.g., Bruhwiler et al., 2014), Geostatistical (e.g., Michalak et al., 2004;Miller et al., 2014). Inversion models also employ different atmospheric transport models and prior (bottom-up) fluxes as constraints. Therefore,

differences in optimisation algorithms/approaches, transport and prior flux errors and their uncertainties can affect the inversion results. Furthermore, differences in observational platforms (e.g. surface measurements, aircraft measurements, remote sensing measurements) and the limited observational information also have impacts on the optimisation of the $CH_4$ fluxes.

Canada has a large Arctic/sub-Arctic region with wetland and permafrost. It is important to study the

methane cycle and monitor the impact of climate change in this sensitive region as it impacts atmospheric $CH_4$ levels at national, continental and hemispheric scale. Environment and Climate Change Canada (ECCC) has recently added five GHG measurement sites in the north to monitor the time evolution of Arctic GHG and to help constrain



flux estimates in the region. In October 2010, ECCC started the measurement at Behchoko (BCK, 115.9˚W, 62.8˚N) that is the first ground-based site of continuous measurement in the Canadian Arctic (representing the land region of Canada north of 60˚N), except for Alert (ALT, 82.5˚N, 62.5˚W) which started in 1978. Following BCK, more continuous measurement systems have been installed, Churchill (CHL, 82.5˚N, 62.5˚W) in 2011, Inuvik (INU,

68.3˚N, 133.5˚W) and Cambridge Bay (CBY, 69.1˚N, 105.1˚W) in 2012. The most recent one is at Baker Lake (BLK, 64.3˚N, 96.0˚W) in July 2017.

This is the first study to present and analyse the atmospheric $CH_4$ concentrations from the new observation sites in the Canadian Arctic region, and to use the observational information in a regional Bayesian inversion framework to infer the Arctic region $CH_4$ fluxes. Then the possible linkage of $CH_4$ fluxes with

climate/environmental variations is examined. The description of the measurement stations as well as the observational data analyses from daily to inter-annual time scales are given in Section 2. The inversion model framework is described in Section 3, and the results are presented and discussed in Section 4.

## 2 Measurements

ECCC has been operating six measurement sites around the Canadian Arctic region to monitor the GHG

concentrations. Alert (ALT) is the most northern GHG monitoring site on the globe since the research laboratory was established in 1988. The other five arctic/sub-arctic sites, Behchoko (BCK), Churchill (CHL), Inuvik (INU), Cambridge Bay (CBY), and Baker Lake (BLK), have become operational gradually since 2007. BLK is the newest site in the Canadian Arctic; the flask air sampling measurement program began in 2014, and continuous measurement started in July 2017. At the four other sites, continuous measurement systems were installed during

the period of 2010–2012, and these observational data were used for the inversion in this study. The information of the six measurement sites are in Table 1, and their locations are shown in Fig. 1. Currently, all the ECCC continuous measurements are performed using an in-situ cavity ring-down spectrometer (CRDS, Picarro G1301, G2301 or G2401), and discrete flask air sampling measurements are performed using a gas chromatograph equipped with flame ionisation detectors (GC-FID, Agilent 6890). Both measurements are calibrated against the World

Meteorological Organization (WMO) X2004 scale (Dlugokencky et al., 2005). In the following sections, we describe the sites briefly and characterise the observed variations of the $CH_4$ concentrations at the sites.

### 2.1 Site Descriptions

**Alert (ALT, 82.5⁰N, 62.5⁰W)** has been referred to as an Arctic background site, being located thousands of kilometres from major source regions. The Alert observatory is ~6 km away from the military base camp. The lack

of local source surrounding the site results in no significant diurnal variation in observed atmospheric $CH_4$ concentrations all year around. In the winter, under weak vertical mixing, well-defined synoptic variations are observed due to inter-continental scale transport along with mainly anthropogenic $CH_4$ originating from the Eurasia continent (Worthy et al., 2009). The measurements at Alert can represent the large-scale background conditions like long-term trend and mean seasonal cycle in the Arctic (Worthy et al., 2009).




**Behchoko (BCK, 115.9˚W, 62.8˚N)** is located on the northwest tip of Great Slave Lake. The continuous measurement was started in October 2012. There is no flask sampling at this site. The air sampling intake is at the top of a 60 m communication tower in a local power generation station; 10 km away from the town of Behchoko, a community ~80 km northwest of Yellowknife, the capital of Northwest Territories. Mixed forests, lakes and ponds
surround BCK.

**Inuvik (INU, 133.5˚W, 68.3˚N)** is ~120 km south of the coast of the Arctic Ocean. The continuous measurement was started in February 2012, followed by flask sampling in May 2012. The measurement system is located in the ECCC upper air weather station building, 5 km southeast of the town of Inuvik. INU is ecologically surrounded by
Arctic tundra and geologically located in the east channel of the Mackenzie Delta where a number of water streams and ponds are formed, and vast hydrocarbon deposits are found. Although there are proposed developments of natural gas and pipeline project, they have been on hold.

**Cambridge Bay (CBY, 105.1˚W, 69.1˚N)** is on the southeast coast of Victoria Island. CBY is located ~1 km north
of the town of Cambridge Bay, the largest port of the Arctic Ocean's Northwest Passage. Both continuous and flask sampling measurements were started in December 2012.

**Baker Lake (BKL, 96.0˚W, 64.3˚N)** is on the shore of Baker Lake, ~320 km inland of Hudson Bay. Weekly flask air sampling has been conducted since June 2014, and the continuous measurement was started in July 2017. The air
sampling system is located in the ECCC upper air weather station.   As same with INU, BCK is in the midst of Arctic tundra and small lakes.

**Churchill (CHL, 93.8˚W, 58.7˚N)** is located on the west coast of Hudson Bay. The GHG monitoring program began with flask air sampling in 2007 before the continuous measurement was initiated in October 2011. The
sampling equipment is installed in the Churchill Northern Studies Research Facility, ~23 km east of the town of Churchill. CHL is situated with Arctic tundra to the north and in the northern perimeter of Hudson Bay Lowland, the largest boreal wetland in North America.

### 2.2 Temporal Variations

Figure 2 shows the time-series of $CH_4$ concentrations; the hourly-means and their afternoon means (between
12:00─16:00 local time) from continuous measurements, and from flask sampling. The fitted curve and long-term trend to the merged time-series of afternoon mean continuous measurements and flask sampling measurements at each site are also plotted. The curve-fitting method applied to all the merged time series has two harmonics of one-year and a half-year cycles and two low and high pass digital filters with cut-off periods of 4 months and 24 months respectively (Nakazawa et al., 1997).
35          Overall the features of the continuous and flask measurements are similar regarding long-term trend and seasonal cycle.  Compared to the weekly flask sampling measurements, continuous measurements reveal short





timescale variations. The diurnal and synoptic concentration variations are indications of local and regional scale interactions between the atmosphere and the source fluxes (Chan et al., 2004).

All the sites show similar upward trends of atmospheric $CH_4$. The growth rates at the Canadian Arctic sites are comparable to the global mean growth provided by NOAA based on the global network (Fig. S1). In 2014, the growth rates jumped at all the sites except BCK. In the following year 2015, the growth rates were lowered, but still higher than the ones prior to 2014. The rapid enhancement in growth rates at the Canadian Arctic sites is consistent with the globally averaged atmospheric $CH_4$ (www.esrl.noaa.gov/gmd/ccgg/trends_ch4/). The 2014 growth rate at BCK was also enhanced, but the enhancement was not as high as the other Arctic sites. This moderate growth rate for BCK might be an artefact in its long-term component partially due to a data missing period for two months (mid-November, 2014 to mid-January 2015).

### 2.2.1 Seasonal and inter-annual variations

Since the long-term trends reflect the global-scale source/sink changes, the long-term component at ALT is subtracted from all the sites to focus on the regional scale features in the observed atmospheric $CH_4$ data (Fig. 3). The mean seasonal cycles are high in winter and low in summer. The summer minimum is mainly due to strong chemical reaction with OH in the warm season. All are relatively in phase in winter with high peaks around January/February, while the site differences are more noticeable in summer. The summer minimum representative of the large-scale Arctic background condition evident at Alert occurs in July to August. The summer minima at the other Arctic sites could vary considerably as they are the superposition of the enhanced $CH_4$ sink and increased wetland emissions during warmer seasons. Minima are seen in June at BCK, INU and CHL, followed by BKL and CBY with ~1 to 1.5 month lags. In fact, INU, BCK and CHL have a summer secondary maximum feature (a summer bump), indicative of the influence of local/regional wetland and biomass burning emissions. As seen in Figure 3, these summer bumps are not regular in timing and amplitude, but vary year-to-year. The bumps were observed at BCK, INU and CHL in 2012 which were in phase with each other, but not at any site in 2013. In 2014, a larger summer bump was observed at BCK than 2012 (2014 has strong biomass burning contributions) while the summer bump at CHL was similar to the one in 2012. The cause(s) for the "summer bumps" at BCK, CHL and INU might vary year-to-year, such as local/regional (wetland and forest fires) emission change due to climate anomaly. Other possible cause is Inter-annually varying atmospheric transport.

### 2.2.2 Synoptic and diurnal variability

All measurements of atmospheric $CH_4$ in the Canadian Arctic show synoptic and daily variations with seasonally changing amplitudes. One quantitative measure of synoptic variability in the observed $CH_4$ concentrations is the monthly Standard Deviation (SD) of the observed time series to their fitted curves. Figure 4 shows the mean seasonality in SD of all 24 hourly data (SD_24) in $CH_4$ concentrations at each site except BKL, as well as the mean seasonality of the afternoon hourly data (SD_PM). Although SD_24 and SD_PM appear similar (some are almost identical) except during the summer months, the differences between SD_PM and SD_24 give a measure if the daily



variability is reflecting a local scale change in emission or rather seasonally changing atmospheric transport processes.

The most substantial synoptic variations are observed in summer at all sites except ALT (Fig. 4). This indicates that the major regional $CH_4$ emissions in the continental Canadian Arctic occur in summer. In winter, the

largest synoptic variation is observed at ALT. The synoptic variations are relatively large for the rest of the sites. The wintertime variability might indicate local anthropogenic emission signals rectified under the winter shallow planetary boundary layer (PBL) or strong long-range transport from other regions which has been demonstrated for ALT (Worthy et al., 2009).

The diurnal variability of atmospheric $CH_4$ is mainly caused by a local $CH_4$ emission signal modulated by

daily PBL development, or a temporal change of the local source. In the summer, the SD_24 values are higher by > 5ppb than the SD_PM except for ALT. The larger SD_24 in the summer supports the existence of local $CH_4$ sources around the sites, likely wetland $CH_4$ emissions. In contrast, the fact ALT has identical SD_24 and SD_PM all year round confirms that there is no significant local source at ALT as mentioned earlier.

Like the three continental sites (BCK, INU, CHL), CBY also shows the maxima of SD_24 and SD_PM in

summer, but they remain lower than BCK, INU and CHL, but higher than ALT. This indicates that weaker local source of $CH_4$ around CBY than the three continental sites. In the cold season (September to May), the SD_24 and SD_PM at CBY are almost identical to ALT. It is noticeable that the SD_24 and SD_PM at BCK, INU and CHL are still higher than ALT until December. These higher SD_24 and SD_PM values in the first half of the cold season might indicate the $CH_4$ emissions from the ground. Zona et al. (2016) suggested the $CH_4$ emissions from the

Alaskan Arctic tundra during the "zero curtain" period when the soil temperature is near zero with average air temperature below 0°C until the surface is completely frozen.

The SD_24 and SD_PM for winter to spring (January to May) at INU remain higher than the other sites. Also, SD_24 at INU becomes higher than SD_PM from April, and remains higher over summer. At the other sites, the difference between SD_24 and SD_PM are seen mainly in summer months (June–August). This higher

variability in atmospheric $CH_4$ at INU in winter and spring, when the surrounding wetland ecosystem is inactive, might indicate a strong local $CH_4$ source, such as anthropogenic $CH_4$ emission from natural gas well/refinery facilities. During the winter, such local $CH_4$ signals are amplified by the seasonally calm condition (the mean seasonal cycles of wind speed are shown in Fig. S2 in the Supplemental Information Section) as well as by less vertical mixing under the shallow PBL due to limited winter daytime in the polar region. Figure S3 shows the

deviations (from the fitted curve) of observed hourly and afternoon mean $CH_4$ at INU and BCK along with the wind speed. For April and May, the difference of deviations between hourly $CH_4$ and afternoon mean $CH_4$ becomes larger again after the relatively quiet period. This could indicate the signals of local (anthropogenic) emission around INU are amplified as the PBL diurnal variation starts developing due to longer daytime. Another possible local source for the large spring SD_24 and SD_PM at INU is natural $CH_4$ emission from lakes and ponds during the

spring thaw (Jammet et al., 2015). In contrast, SD_24 and SD_PM at BCK become less as the wind speed becomes higher, indicating a lack of local $CH_4$ source around BCK in spring.





Since ALT is representative of the Arctic background state in synoptic variability, the difference of SD_24 or SD_PM between ALT and each of other sites gives a measure of the regional source influence to the site. The large regional source influence signals in summer shown in Figure 4 should be useful in constraining the regional flux estimation modelling in the next sections.

**3 Model description**

To estimate the regional $CH_4$ fluxes in the Canadian Arctic, we apply a Bayesian inversion approach, based on the backward simulations by Lagrangian Particle Dispersion Models (LPDM). In this study, three different transport models and three prior $CH_4$ flux distributions were used to help estimate the model uncertainties. The following sections describe the various components of our regional inverse modelling.

**3.1 Transport models and meteorological data**

LPDMs simulate an ensemble of air-following particles which are released from the measurement sites. The air particles travel backwards in time for 5 days with the wind field. Previous studies (e.g. Cooper et al., 2010; Gloor et al, 2001; Stohl et al, 2009) have shown 5 days are typically sufficient to capture the surface influence to a measurement site from the surrounding region. The backward trajectory is used to calculate the footprints as the

integrated residence times the particles spent inside the PBL at a resolution of 1.0°×1.0. We use three different regional model settings combining two different LPDMs: FLEXPART and STILT, and three different meteorological data from the European Centre for Medium-range Weather Forecasts (ECMWF), Japanese Meteorological Agency (JMA), and Weather Research and Forecasting model (WRF).

LPDMs simulate local contributions for 5 days prior to the measurements at sites. The background

condition of atmospheric $CH_4$ concentrations at the endpoints of the particles is provided by a global model, National Institute for Environmental Studies-Transport Model (NIES-TM) with global $CH_4$ flux fields. Below are the details of model settings in this study.

**3.1.1 LPDM: FLEXPART_EI**

The first model setting is FLEXible PARTicle dispersion model (FLEXPART) (Stohl et al., 2005) driven by

Reanalysis meteorology from the European Centre for Medium-range Weather Forecasts (ECMWF) ERA-Interim (Dee et al., 2011; Uppala et al., 2005). The input meteorological data are at 3-hourly time step and interpolated to 1.0°×1.0° horizontal resolution with 62 vertical layers.

**3.1.2 LPDM: FLEXPART_JRA55**

The second model setting is also FLEXPART, but driven by the Japanese 55-year Reanalysis (JRA55) from

Japanese Meteorological Agency (JMA, Kobayashi et al., 2015; Harada et al., 2016). JRA55 is at 6 hourly time step resolution and on TL319 (~0.5625˚, ~55 km) horizontal resolution and, has 60 vertical layers. For this study, we used the JRA55 dataset at half the resolution (~1.25˚). This model setting was used for a global inverse modelling



system by Global Eulerian-Lagrangian Coupled Atmospheric Model (GELCA). GELCA is a coupled atmospheric model of NIES-TM and FLEXPART (Ishizawa et al., 2016). The primary meteorological observational data for JRA55 have been supplied by ECMWF. In addition to the ECMWF data, the observational data obtained by JMA and other sources are also used.

### 2.1.3 LPDM:WRF-STILT

The third model setting uses Stochastic, Time-Inverted, Lagrangian Transport Model (STILT) (Lin et al., 2003; Lin and Gerbig, 2005). The wind fields to drive STILT are from the Weather Research and Forecasting model (WRF) (Skamarock et al., 2008) on 10 km resolutions. Detailed descriptions are found elsewhere (Hu et al., 2018; Miller et

al., 2014; Henderson et al., 2015). The footprints are aggregated to $1.0° \times 1.0°$ horizontal resolution, similar to the other models in this study. The STILT footprint data are provided from CarbonTraker Lagrange which is a Lagrangian assimilation framework developed at NOAA Earth System Research Laboratory.

### 3.1.4 Global background model: NIES-TM

The background or initial condition for the LPDMs is obtained by sampling a global model of $CH_4$ at the 5-day back

endpoint locations of the LPDM particles. The global background concentration field is simulated by NIES-TM version 8.1i (Belikov et al., 2013) with the optimised $CH_4$ fluxes with GELCA-$CH_4$ inversion system (Ishizawa et al., 2016; Saunois et al., 2016). The GELCA-$CH_4$ inverse modelling system optimised the monthly $CH_4$ fluxes for 2000-2015 to assimilate a global network of surface $CH_4$ measurements available through GAW World Data Center for Greenhouse Gases (WDCGG, http://ds.data.jma.go.jp/gmd/wdcgg). The prior $CH_4$ fluxes for the GELCA-$CH_4$

global inversion are also used for the regional inversion in this study as described in the later section. The NIES-TM has $2.5° \times 2.5°$ horizontal resolution and 32 vertical layers, driven by JRA55. For the global simulation, the $CH_4$ loss in atmosphere is included; the stratospheric $CH_4$ loss and OH oxidation schemes are adapted from a model inter-comparison project "TransCom-$CH_4$" (Patra et al., 2011).

### 3.2 Prior fluxes

Three cases of prior emissions, C1, C2 and C3, were used as listed in Table 2. C1 and C2 are from the prior and posterior fluxes for the global inversion by GELCA, respectively. In this study, the mean fluxes for the last 5 years of the GELCA global inversion were used. C3 is the same set with C2, but wetland $CH_4$ fluxes are from WetCHARTs (a global wetland methane emission model ensemble for use in atmospheric chemical transport models) which are inter-annually varying. The details are described in the following sections.

### 3.2.1 Wetland $CH_4$ fluxes

We used the monthly $CH_4$ wetland fluxes from two different models. The first model is Vegetation Integrative Simulator for Trace gases (VISIT) (Ito and Inatomi, 2012). VISIT is a process-based model, using GLWD as wetland extent. Beside wetland $CH_4$ flux, VISIT calculates soil $CH_4$ uptake and $CH_4$ emission through rice



cultivation. The wetland fluxes combined with $CH_4$ fluxes from rice cultivation were optimised through the GELCA-$CH_4$ global inversion as a natural $CH_4$ flux. The second model is WetCHARTs version 1.0 (Bloom et al., 2017a). WetCHARTs derives wetland $CH_4$ fluxes as a function of a global scaling factor, wetland extent, carbon heterotrophic respiration and temperature dependence (Bloom et al., 2017b). We used the ensemble mean fluxes

over 18 model sets which are available for 2001-2015, using 1) three global scaling factors, 2) two wetland extents: GLWD and GLOBCOVER, 3) CARDAMOM (the global CARbon Data MOdel fraMework) as terrestrial carbon analysis, and 4) three temperature dependent $CH_4$ respiration functions. The WetCHARTs horizontal resolution is $0.5° \times 0.5°$. The modelled $CH_4$ fluxes are aggregated into $1.0° \times 1.0°$ for this study. Figure 5 shows the spatial distribution of three wetland $CH_4$ fluxes for the summer months (July-August). Overall they are similar, while

WetCHARTs (C3) has stronger emissions in Northwest Territories than the two wetland fluxes from VISIT (C1 and C2).

### 3.2.2 Forest fire $CH_4$ fluxes

GFAS (Global Fire Assimilation System) v1.2 (Kaiser et al., 2012) provides biomass burning (BB) emissions by assimilating Fire Radiative Power (FRP) from the Moderate resolution imaging Spectrometer (MODIS). The FRP

observations are firstly corrected for data gaps and then linked to dry matter combustion rates with $CH_4$ emission factors. GFAS has a daily temporal resolution and $0.1° \times 0.1°$ horizontal resolution. In this study, the daily fire $CH_4$ emissions are spatially aggregated into $1.0° \times 1.0°$ resolutions for the regional inversion, though monthly fluxes were used for the GELCA global inversion.

### 3.2.3 Anthropogenic Emission

The anthropogenic $CH_4$ emissions are provided from EDGAR (Emission Database for Global Atmospheric Research) v4.2FT2010 (http://edgar.jrc.ec.europa.eu), except for rice cultivation. EDGARv4.2FT2010 emission which is originally at $0.1° \times 0.1°$ resolution is aggregated into $1.0° \times 1.0°$. Since the EDGARv4.2FT2010 data are available until 2010, the same values for 2010 are used for the years beyond 2010. The $CH_4$ emission from rice cultivation was replaced with the one from VISIT-$CH_4$ and then treated as a part of natural fluxes because there is

no rice field in the Canadian Arctic and also in the rest of North American Arctic/Boreal region, the influence of $CH_4$ emission from rice cultivation in the region of interest in this study is negligible. The difference of the optimised anthropogenic emissions in the Canadian Arctic from the prior by the global GELCA inversion is almost negligible (from 0.0247 $TgCH_4$ $yr^{-1}$ to 0.0250 $TgCH_4$ $yr^{-1}$). Compared to the wetland emissions, the emissions are substantially smaller and localised (see Fig. 5).

### 3.2.4 Other natural $CH_4$ fluxes

For other natural $CH_4$ fluxes, we used a climatological emission map of termite from Fung et al. (1991) and modelled soil uptake from VISIT-$CH_4$. Because of no termite $CH_4$ emissions in the Canadian Arctic, termite $CH_4$ emission has no direct impact, but it is included in global simulation for the background concentration. The prior soil $CH_4$ uptake is provided by VISIT-$CH_4$ as oxidative consumption by methanotrophic bacteria in unsaturated





lands. Soil $CH_4$ uptake has large uncertainty regionally and also globally. Kirschke et al. (2013) reported that the global soil uptake ranges from 9 to 47 $TgCH_4\,yr^{-1}$. In the Canadian Arctic, the VISIT-modelled soil uptake is weak (0.094 $TgCH_4\,yr^{-1}$) but spread widely (Fig. 5). In some parts of the eastern Canadian Arctic, soil uptakes exceed other $CH_4$ emissions, resulting in negative fluxes/net sink of atmospheric $CH_4$.

### 3.3 Inversion Setup

#### 3.3.1 Regional inversion

In this study, we use the Bayesian Inversion approach. The Bayesian inversion optimises the scaling factors of posterior fluxes by minimising the mismatch between modelled and observed concentrations with constraints and given uncertainties using the cost function (J) minimisation method (Lin et al., 2004).

$$J(\lambda) = (y - K\lambda)^T D_\epsilon^{-1}(y - K\lambda) + (\lambda - \lambda_{prior})^T D_{prior}^{-1}(\lambda - \lambda_{prior}) \tag{1}$$

where $y$ (N×1) is the vector of observations (with the background concentration representing the modelled $CH_4$ signal from 5 days prior to the observation time subtracted, see Section 3.2.4), N is the number of time points times number of stations (N is reduced if observations are missing). $\lambda$ (R×1) is the vector of the posterior scaling factors to be estimated, R is the number of sub-regions to be solved. $\lambda_{prior}$ is the vector of the prior scaling factors which are all initialised to 1 for all sub-regions, and $K$ (N×R) is the matrix of contributions from R sub-regions. $K$ is a Jacobian matrix of flux sensitivity, a product of two matrices, $M$ and $x$. $M$ is the modelled transport (or footprints in this study), and $x$ is the spatial distribution of the surface fluxes. A linear regularisation term has been added which is the second term on the right-hand side of the equation. $D_\epsilon$ and $D_{prior}$ are the error covariance matrices. $D_\epsilon$ is the prior model-observation error/uncertainty matrix (N×N) where the diagonal elements are $(\sigma_e)^2$. $D_{prior}$ is the prior scaling factor uncertainty matrix (R×R) where the diagonal elements are $(\sigma_{prior})^2$. We assume that the model-observation mismatch errors are uncorrelated each other and the contributions from the sub-regions are uncorrelated. All the off-diagonal elements in $D_\epsilon$ and $D_{prior}$ are assumed to be zero. We assigned $\sigma_e$= 0.33 for the model-observation error and $\sigma_{prior}$= 0.30 for the prior uncertainty. We examined the inversion's sensitivity to these uncertainties by doubling their values. The results showed the optimised fluxes are not strongly dependent on these prescribed uncertainties. The estimate for $\lambda$ is calculated according to the expression below (Lin et al., 2004).

$$\lambda = \left(K^T D_\epsilon^{-1} K + D_{prior}^{-1}\right)^{-1}\left(K^T D_\epsilon^{-1} y + D_{prior}^{-1}\lambda_{prior}\right) \tag{2}$$

The posterior error variance-covariance, $\Sigma_{post}$, for the estimates of $\lambda$ is calculated,

$$\Sigma_{post} = \left(K^T D_\epsilon^{-1} K + D_{prior}^{-1}\right)^{-1}. \tag{3}$$

We optimise the $CH_4$ fluxes from biomass burning and separately the remaining fluxes (consisting of wetland emission, soil uptake and anthropogenic emission) on a monthly time resolution.



### 3.3.2 Domain/Sub-regions

We set up three sub-region masks for the Canadian Arctic based on three territories 1) Northwest Territories (NT), Yukon (YT), and Nunavut (NU) as shown in Fig. S4. Outside of the Canadian Arctic is treated as one outer region. Regarding the subdivision of the Arctic region, we examined the sensitivity of the flux estimation to the number of

sub-regions. As a starting point, the three territories are treated separately. Secondly, YT is combined with NT. There is no existing measurement site in Yukon and no significant $CH_4$ emissions in prior fluxes. The inversion results in the next section will show YT could not be reliably constrained as a separate sub-region. As the third region mask, we solve the fluxes for one region representing the entire Canadian Arctic. Like YT, NU is a weak source region, compared to NT, and weak observational constraint might lead to unrealistic flux estimates. This

exercise on the subdivision gives insights on the constraining power of the existing measurements. Table 3 shows all the inversion experiments in this study. We perform totally 27 experiments with 3 prior emission cases, 3 different transport models and 3 different sub-region masks.

### 3.3.3 Atmospheric Measurements

This regional inversion study used the continuous measurements at BCK, INU, CBY, and CHL for the four years,

2012─2015 (Fig. 2). Firstly the afternoon mean values are calculated by averaging the hourly data over 4 hours from 12:00 to 16:00 local time, and then the modelled background concentrations, which were described earlier, are subtracted from the afternoon means. The concentration differences between observed and background concentrations were input into the regional inversion system as local contributions. The observational data examined in Section 2 have been already pre-screened for possible contaminations due to mechanical/technical

problems during sampling /analysing processes. Except for the pre-screening, we did not apply any additional data screening or filtering.

## 4 Results and Discussions

### 4.1 Comparison of footprints

Figure 6 shows the mean footprints (mean emission sensitivities) of all 4 sites by the three different LPDMs. There

are common features, but there are also noticeable seasonal differences and differences between the models. The spatial coverage is similar, but the sensitivity to emissions around sites depends on the models. Among the models, STILT shows the strongest sensitivity near the sites, while FLEXPART_JRA55 has the weakest sensitivity. All the footprints near the sites for the winter season are stronger than the summer season. The footprint differences among the models are also more significant. STILT appears to be more localised to the sites. These differences indicate that

choosing multiple implementations for the atmospheric transport will allow us to reflect some of the uncertainties introduced to our inversion estimate by transport models.



### 4.2 Signals in the observations (relative to background)

The regional inversion depends on how well local signals can be detected in the observations. Therefore, we first look at the detectability of local/regional fluxes in the observed atmospheric $CH_4$ concentrations. If the amplitude of local signals is comparable to the background contribution, estimated regional fluxes would be more uncertain because local signals would be difficult to distinguish from the background contributions. In Section 2, we examined the synoptic variability in observed $CH_4$ concentrations. Here we apply the same procedures to the modelled background concentrations for the sites to see if the local synoptic signal is distinguishable from the background concentrations. Figure 7 shows the mean monthly SD of modelled background $CH_4$ concentrations to their fitted curves for the case of FLEXPART_EI (other model settings are analogous), along with those of observed $CH_4$ concentrations (SD_PM in Fig. 4) for the 4 sites to be used as observational constraints (BCK, INU, CBY and CHL). In summer, all the SD_PM values of the observations are much larger (up to three times), than the respective background SDs, indicating strong local influence. While, in winter, both the observation SD_PM and the background SD are comparable. Thus, the observations could provide more constraints on the estimated regional fluxes in summer than in winter.

### 4.3 Comparison of prior and posterior fluxes with different transport models

The inversion experiments outlined in Table 3 were done to estimate the $CH_4$ fluxes in the Canadian Arctic using atmospheric observations from the aforementioned five ECCC stations. We calculated the posterior flux estimates as the mean of the fluxes estimated in the 9 experiments in Table 3 (for each set of sub-region masks). The variations in the flux results (Standard Deviation) are used to represent the flux uncertainty due to transport errors (3 transport models) and prior flux errors (3 prior emission cases). This flux uncertainty is larger than the posterior flux covariance uncertainty estimates, Eq. (3). Figure 8 shows the monthly posterior fluxes with sub-region masks A and B. The monthly posterior fluxes with mask C are showed in Fig. S5, along with the aggregated fluxes with masks A and B for the entire Canadian Arctic. As shown in Fig.8a, the fluxes in NT are dominant, and all the posterior fluxes in NT show clear seasonal cycle and inter-annual variations that are reflected in the total fluxes for the entire Canadian Arctic (Fig. S5). In contrast, no clear seasonal pattern is found for NU and YT (Figs. 8a and 8b). The inversion model has difficulty optimising the weak flux regions. As a result, negative mean fluxes, i.e. $CH_4$ sinks, could appear, especially in YT (Fig. 8a); however a null-flux would be consistent within error bars.

Next, the differences and similarities in the inversion results from the three transport models are summarised. The differences in the flux estimates by the three different transport models can be seen in Fig. 9. Figure 9 displays the example of the experiments with Mask B by the three different transport models for YT+NT. FLEXPART_JRA55 tends to estimate higher total fluxes than the other models, resulting in higher emissions by ~0.6 $TgCH_4$ yr$^{-1}$ than the average of ~1.8 $TgCH_4$ yr$^{-1}$. WRF-STILT tends to yield the lowest estimate among the three models, lower by ~0.5 $TgCH_4$ yr$^{-1}$ than the average. The posterior total fluxes by FLEXPART_EI appear to be moderate. In the winter, the FLEXPART-EI fluxes are close to zero, same with WRF-STILT. These results are consistent with their footprints (mean emission sensitivities) in Fig. 6. Higher footprint sensitivities near the sites tend to yield lower posterior fluxes and vice versa.



The inter-model differences in the posterior forest fire fluxes (Biomass Burning, BB) are quite significant in 2014 that is the extreme fire year in NT. Due to the sporadic nature of the fire events, the differences in transport (transport errors) are evident in the modelled prior concentrations (Fig. S6c) and could lead to substantial differences in the posterior fluxes (Fig. 9). The WRF-STILT estimated BB in 2014 appears to be moderate ($0.23$ TgCH$_4$ yr$^{-1}$), similar to in 2013 (~$0.3$ TgCH$_4$ yr$^{-1}$), while the other two models show the highest BB flux estimates ($0.55$−$0.67$ TgCH$_4$ yr$^{-1}$) in 2014, comparable to the prior flux, GFAS estimates.

In contrast, the inter-annual variability in total posterior fluxes is very similar among all three transport model results (as shown in Figs. 8 and S5). The inter-annual variability in the transport models (an intra-model result) appears to be consistent, yielding similar posterior flux inter-annual variability. Since all three different transport models capture this inter-annual variability, it appears to be a robust feature of the CH$_4$ source/sink in the Canadian Arctic.

Another robust feature appears to be the similarity in the results for the total Arctic emission with different numbers of sub-regions used in the inversion. The sub-region with strong signals in the prior fluxes (NT) and strong observational constraints (BCK and INU within NT) yielded posterior flux results with small uncertainties, while sub-region with weak signals in the prior fluxes (YT) and weak observational constraint (no observations in YT) yielded large uncertainties in the posterior flux estimates. Weak sub-region like YT could be combined with other sub-region (NT) without strong impact on the inversion results. The temporal variations in the inversion results with different numbers of sub-regions (an intra-model result) seem to be a robust feature also. Given that the strong observational constraints and the strong wetland emissions are both located in the central part of the Canadian Arctic, representing the Canadian Arctic as a single region was able to yield reasonable inversion results.

## 4.4 Comparison with previous estimates

The estimated fluxes for the entire Canadian Arctic in this study are relatively robust in amplitude and temporal variations even with the different prior fluxes and sub-region masking. The mean estimated total CH$_4$ annual flux for the Canadian Arctic is $1.8 \pm 0.6$ TgCH$_4$ yr$^{-1}$. Compared with two previous inversion estimates, our estimate is slightly lower than the mean total flux of $2.14$ TgCH$_4$ yr$^{-1}$ (average from 2009−2013) inferred by FLEXINVERT regional inversion (Thompson et al. 2017), but much higher than the estimate of $0.5$ TgCH$_4$ yr$^{-1}$ (average from 2006−2010) from the CarbonTracker-CH$_4$ global inversion (Bruhwiler et al., 2014) (Fig. 10 a).

All the estimated fluxes are seasonally high around July and August (Fig. 10b). The mean summertime maximum of our estimates is quite consistent with the one by Thompson et al. (2017), but our estimated fluxes have narrow high summer emission period and low wintertime emission compared with the estimates by Thompson et al. (2017). These temporal differences in estimated fluxes might reflect the observational constraints used in the respective inversions. Thompson et al. (2017) employed a similar type of regional inversion but for the entire northern high latitudes (north of 50˚N). Except for the flask measurement data at CHL, none of the Canadian Arctic sites used in this study was included in Thompson et al. (2017). The strong regional CH$_4$ signals at INU and BCK in this study appear to yield flux estimates with narrower high summer emission period and lower wintertime wetland emission compared with the estimates by Thompson et al. (2017).



The flux estimate in this study is partitioned into biomass burning (BB), $0.3 \pm 0.1$ TgCH$_4$ yr$^{-1}$, and the remaining flux $1.5 \pm 0.5$ TgCH$_4$ yr$^{-1}$. The remaining flux is mainly natural/wetland CH$_4$ emissions, given that anthropogenic contribution to the total prior fluxes without BB is ~2 % according to the EDGAR prior fluxes. The estimated wetland flux is comparable to the WetCHARTs (wetland) ensemble mean of 1.35 TgCH$_4$ yr$^{-1}$ (Bloom et al. 2017a; 2017b).

The estimated summertime natural CH$_4$ fluxes show clear inter-annual variability. The higher emissions are estimated in 2012 and 2014 in this study, which is similar to the results from Arctic Reservoirs Vulnerability Experiment (CARVE) aircraft measurements over Alaska for 2012 to 2014 (Hartery at al., 2018).

**4.5 Comparison of prior and posterior concentrations to observations**

The model-observation statistical comparison is shown with the Taylor diagrams of correlation coefficients and normalised standard deviation (NSD) by three different transport models for the four Arctic sites (Fig. 11) using the inversion results with Mask B and prior flux case C3. At BCK and INU, the correlation coefficients and NSD for each model are improved by the inversion. At these two sites, the observations contain large synoptic signals from the Canadian Arctic wetland and provide strong constraints to the inversions. At INU, the improvement for STILT is noticeable, especially with NSD. This is explained further below. At CBY and CHL, no significant changes between the prior and posterior results are seen. This indicates that the regional flux in the Canadian Arctic only weakly influences CBY and CHL.

Further investigation has been done for INU. Figure S7a shows the time-series of modelled concentrations by the three transport models and the observed concentrations. The Taylor diagrams in Fig. S7b show the results annually and by seasons, summer months (June─September) and winter months (October─May) separately as well as for the entire period together. The modelled concentrations by STILT with the prior fluxes could be much higher than the concentrations by the other two models. That results in the higher prior NSD values, especially in winter season. The inversion was able to improve the results by reducing the fluxes and consequently the posterior NSD.

**4.6 Sensitivity test**

**4.6.1 Prior fluxes: wetland CH$_4$ fluxes**

Wetland CH$_4$ emissions are the dominant flux in the Canadian Arctic. To examine how the prior fluxes impact on the posterior fluxes, two inversion experiments were conducted with modified WetCHARTs fluxes. One is 50 % reduced emissions in the Canadian Arctic, and another is 50 % increased emissions in the Canadian Arctic. The results are shown as mean posterior natural fluxes in Fig. S8. Despite the change in wetland prior emissions, all the posterior fluxes are similar to the ones in the control case; the changes in the posterior fluxes are less than 5% annually. This indicates that the posterior fluxes are not very sensitive to the amplitude/strength of prior fluxes.



### 4.6.2 Contributions of background concentrations on the posterior fluxes

We used the same background contributions for the different transport models, which are calculated using the particle endpoints from FLEXPART_JRA55. The idea of using the same background concentration is to focus on the impact of local/regional transport contribution on regional inversion, separating from the background
contribution.

One notable feature in the background concentrations is the relatively large synoptic variability, especially in winter against the observed concentrations, which might have large uncertainties in flux estimation. To examine how sensitive the inversion results are to these temporal variations in the background concentrations, additional experiments with background concentrations with smoothing windows of 5 days, 10 days and 30 days were done
(see Fig. S9a).

The examples of the results are shown in Figs. S9b and S9c. The posterior fluxes are not strongly dependent on the different background concentrations. Compared with unsmoothed background case, slightly small values of NSD are found in the model-observation statistical comparison for summer with 30-day smoothing. It seems there are sufficient observations (signal level) above the background concentrations (noise level) to constrain
the inversion results (Fig. 7).

### 4.6.3 Relationship of fluxes with climate anomalies

The relationships of the posterior fluxes with climate parameters are examined here, specifically with surface air temperature and precipitation from NCEP reanalysis (Kalnay et al., 1996). First, monthly mean values at the sub-regions are calculated to obtain the monthly anomalies from the 4-year means (2012−2015). The temperature and
precipitation anomalies are aggregated to the respective regions, NT, YT and NU. On the regional level, climate anomalies in NT and NU are quite similar, though YT is less similar to NT and NU. YT is mainly covered by mountains with little wetland. Furthermore, NT has the largest wetland extent and most of the forest fire emissions in 2012-2015. Thus, we look into the correlation in monthly anomalies of $CH_4$ fluxes with the summer climate anomalies in NT.

In Fig. 12, the inter-annual variability of wetland $CH_4$ fluxes exhibits a moderate positive correlation with the surface temperature anomaly (r = 0.57) and only weakly correlated with precipitation anomalies (r = 0.13). This indicates that the hotter summer weather condition stimulates the wetland $CH_4$ emission, and precipitation appears to have a less immediate or no direct impact on wetland conditions.

Inter-annual variations of estimated BB $CH_4$ fluxes show a negative correlation with precipitation (r = -
0.47). Also throughout the fire season (June-September), all estimated BB fluxes negatively correlate with precipitation while the prior BB fluxes appear to have no consistent correlations. The inversion results support that dry condition would enhance the forest fire. The estimated BB fluxes show weakly negative correlation with surface temperature (r = -0.38) on mid-summer average, but the monthly correlations are fluctuating from r = -0.47 to r = 0.69 over the fire season. Since the period is limited in this study (2012−2015), these statistical relationships
are still not clear. Also, the relationship of $CH_4$ emissions with climate conditions could be complex and non-linear



(with extreme fires events in some years). More data and analysis are required to characterise the dependence of $CH_4$ fluxes on climate in the Arctic.

## 5 Summary

The Canadian Arctic region is one of the potential enhanced $CH_4$ source regions related to the ongoing global
warming (AMAP, 2015), and earth system models differ in their prediction how the carbon loss there will be split up
between $CO_2$ and $CH_4$ emissions. Even current bottom-up and top-down estimates of the $CH_4$ flux in the region vary
widely.  This study:

1)  analysed the measurements of atmospheric $CH_4$ concentrations that include 5 sites established in the Canadian
Arctic by ECCC, to characterise the observed variations and examine the detectability of regional fluxes. And,

2) estimated the regional fluxes for 4 years (2012−2015) with the continuous observational data of atmospheric
$CH_4$, employing a Bayesian atmospheric inversion method with three different sets of Lagrangian particle dispersion
model and meteorological data (FLEXPART_EI, FLEXPART_JRA55 and WRF-STILT). In addition to the model
variations, inversion experiments included different sub-region masks and prior emissions to investigate their impact
on the estimated fluxes and their uncertainties. We also examined the relationship of the estimated fluxes with the
climate anomalies.

The observational data analysis reveals large synoptic summertime signals in the atmospheric $CH_4$,
indicating strong regional fluxes (most likely wetland and biomass burning $CH_4$ emissions) around Behchoko and
Inuvik in Northwest Territory, the western Canadian Arctic. These observational signals are quite distinct from the
background signals and could be used for inverse flux estimations.  The local $CH_4$ concentration signals also allow
inverse models to optimise biomass burning $CH_4$ flux (emissions due to forest fire), separately from the
remaining/natural $CH_4$ fluxes (including wetland, soil sink and anthropogenic, but mostly due to wetland $CH_4$
emissions).

The inverse flux estimates included three different transport models, three different prior wetland emission
datasets and three sub-region definitions to help quantify the uncertainties in the results. For wetland, a spatially
distributed and slowly varying $CH_4$ source, the transport models could repeatedly sample the sources around each
site and providing sufficient signals for the inversion model to optimise the fluxes. The estimated wetland flux $1.5 \pm 0.5$ $TgCH_4$ $yr^{-1}$ for the entire Canadian Arctic is relatively robust in amplitude and temporal variation.  The estimated
biomass (BB) burning flux is $0.3 \pm 0.1$ $TgCH_4$ $yr^{-1}$ on average, but strongly dependent on the transport models.  The
large point-like BB emissions with strong temporal (daily) variations near Behchoko coupled with the strong
transport model differences near the site yielded very different modelled prior concentrations at the site.
Consequently inferred BB flux estimates have large uncertainty (particularly for 2014).

The estimated mean total $CH_4$ annual flux for the Canadian Arctic is $1.8 \pm 0.6$ $TgCH_4$ $yr^{-1}$. The mean total
flux in this study is comparable to another regional flux inversion result of 2.14 $TgCH_4$ $yr^{-1}$ by Thompson et al.
(2017), but much higher than the global inversion result of 0.5 $TgCH_4$ $yr^{-1}$ by CarbonTracker-$CH_4$ (Bruhwiler et al.,
2014).  The strong regional $CH_4$ signals at INU and BCK appear to yield flux estimates in this study with narrower



high summer emission period and lower wintertime wetland emission compared with the estimates by Thompson et al. (2017).

Clear inter-annual variability is found in all the estimated summertime natural $CH_4$ fluxes for the Canadian Arctic, mostly due to Northwest Territories. These summertime flux variations are positively correlated with the surface temperature anomaly (r = 0.57). This result indicates that the hotter summer weather condition stimulates the wetland $CH_4$ emission. More data and analysis are required to characterise the dependence of $CH_4$ fluxes on climate in the Arctic. In the future, these Arctic measurement sites should help quantify the inter-annual variations and long-term trends in $CH_4$ emissions in the Canadian Arctic.

**Data availability**

The data measured at ECCC sites used in this study are available upon request to Doug Worthy (doug.worthy@canada.ca). The Alert data are also periodically updated to WDCGG, http://ds.data.jma.go.jp/gmd/wdcgg. The model-related data are available upon request to corresponding author.

**Author Contributions**

MI and DC designed the research and prepared the manuscript. MI performed data analysis and inversion experiments. DW led the measurement programs and collected the observational data. MI, DC and EC provided footprints (potential emission sensitivities) information for inversions. All authors contributed to the discussion and interpretation of the results.

**Competing Interest**

Authors declare that they have no conflict of interest.

**Acknowledgments**

We acknowledges CarbonTraker (CT) Lagrange program for providing the WRF-STILT footprint data for our inversion study. CT_Lagrange has been supported by NOAA Climate Program Office's Atmospheric Chemistry, Carbon Cycle, & Climate (AC4) Program and the NASA Carbon Monitoring System. We would like to extend our gratitude to the conscientious care taken by the program technicians, Robert Kessler and Larry Giroux, and IT support provided by Senen Racki. Appreciation is also forwarded to the Northwest Territories Power Corporation (NTPC) for making available their building facility to house our equipment and tower to string our sampling lines. NTPC also provided IT communication to permit access to our equipment and transmission of data.




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



**Table 1. ECCC atmospheric measurements sites in the Canadian Arctic.**

| Site | ID | Latitude [°] | Longitude [°] | Elevation [m] | Sampling height [m] | Start (conti.) Start (flask) |
|------|----|----|----|----|----|----|
| Alert | ALT | 82.5N | 62.5W | 200 | 10 | 1988/01 1999/10 |
| Behchoko[a] | BCK | 62.8N | 115.9W | 160 | 60 | 2010/10 NA |
| Inuvik[a] | INU | 68.3N | 133.5W | 113 | 10 | 2012/02 2012/05 |
| Cambridge Bay[a] | CBY | 69.1N | 105.1W | 35 | 10 | 2012/12 2012/12 |
| Baker Lake | BKL | 64.3N | 96.0W | 95 | 10 | 2014/06 2017/07 |
| Churchill[a] | CHL | 58.7N | 93.8W | 29 | 60 | 2007/05 2011/10 |

[a] The sites are used in the inversion in this study





**Table 2. Three cases of prior emissions.**

| Source | C1 | C2 | C3 |
|---|---|---|---|
| Wetland[1] | VISIT | VISIT (optimized, as Natural[1]) | WetCHARTs Extended (Ver.1.0) |
| Soil Uptake[2] | VISIT | VISIT (optimized, as Soil Uptake[2]) | VISIT (optimized, as Soil Uptake[2]) |
| Anthropogenic[3] (excl. Rice cultivation) | EDGARv4.2FT2010 | EDGARv4.2FT2010 (optimized as Anthropogenic[3]) | EDGARv4.2FT2010 (optimized as Anthropogenic[3]) |
| Biomass Bunning[4] | GFASv1.2 | GFASv1.2 | GFASv1.2 |
| Rice cultivation[1] | VISIT | VISIT (optimized, as Natural[1]) | VISIT (optimized, as Natural[1]) |
| Termites[1] | GISS | GISS (optimized, as Natural[1]) | GISS (optimized, as Natural[1]) |

C1 used the same prior fluxes with those for global GELCA-CH$_4$ inversion except Biomass Burning. GELCA-CH$_4$ inversion optimised CH$_4$ fluxes for 4 source types, 1) natural, 2) soil uptake, 3) anthropogenic and 4) biomass
5  burning, which are also indicated by superscripted numbers. C2 used the posterior fluxes from global GELCA-CH$_4$ inversion. For C1 and C2, five-year mean of each source type was used. S3 used WetCHARTs extended mean fluxes as wetland CH$_4$, while other fluxes were same with C2. For all the scenarios, GFAS v1.2 daily fluxes are used as biomass burning.



**Table 3.** **Experiment configurations.** **Using each of 3 different masks (A, B and C in Fig. S3), 9 inversion runs were conducted with a combination of 3 prior emission cases (C1, C2, C3 on Table 2) and 3 different models (FLEXPART_EI, FLEXPART_JRA55 and WRF-STILT). Totally 27 inversion runs were conducted.**

| Exp | Mask | | | | | |
| --- | --- | --- | --- | --- | --- | --- |
| | Mask A (NT, YT, NU) | | Mask B (NT+NT, NU) | | Mask C (NT+NT+NU) | |
| | Fluxes | Model | Fluxes | Model | Fluxes | Model |
| Exp1 | C1 | FLEXPART_EI | C1 | FLEXPART_EI | C1 | FLEXPART_EI |
| Exp2 | C1 | FLEXPART_JRA55 | C1 | FLEXPART_JRA55 | C1 | FLEXPART_JRA55 |
| Exp3 | C1 | WRF-STILT | C1 | STILT_WRF | C1 | WRF-STILT |
| Exp4 | C2 | FLEXPART_EI | C2 | FLEXPART_EI | C2 | FLEXPART_EI |
| Exp5 | C2 | FLEXPART_JRA55 | C2 | FLEXPART_JRA55 | C2 | FLEXPART_JRA55 |
| Exp6 | C2 | WRF-STILT | C2 | STILT_WRF | C2 | WRF-STILT |
| Exp7 | C3 | FLEXPART_EI | C3 | FLEXPART_EI | C3 | FLEXPART_EI |
| Exp8 | C3 | FLEXPART_JRA55 | C3 | FLEXPART_JRA55 | C3 | FLEXPART_JRA55 |
| Exp9 | C3 | WRF-STILT | C3 | STILT_WRF | C3 | WRF-STILT |



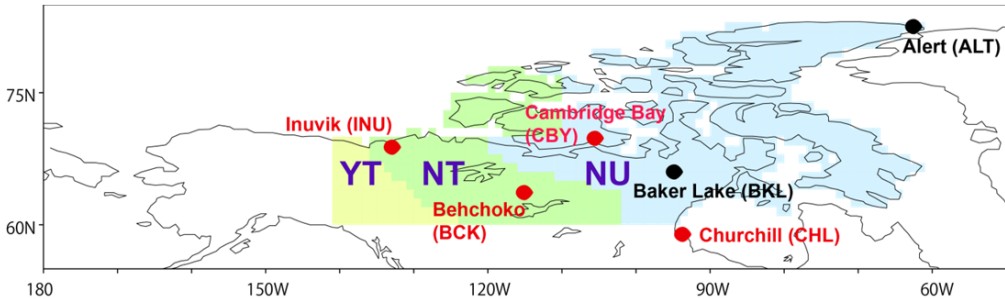

**Figure 1: The ECCC atmospheric measurement sites around the Arctic. The sites used for the inversion are indicated in red. The three shaded areas are the three territories which are used as sub-regions in the inversions: YT (Yukon), NT (Northwest Territories) and NU (Nunavut).**




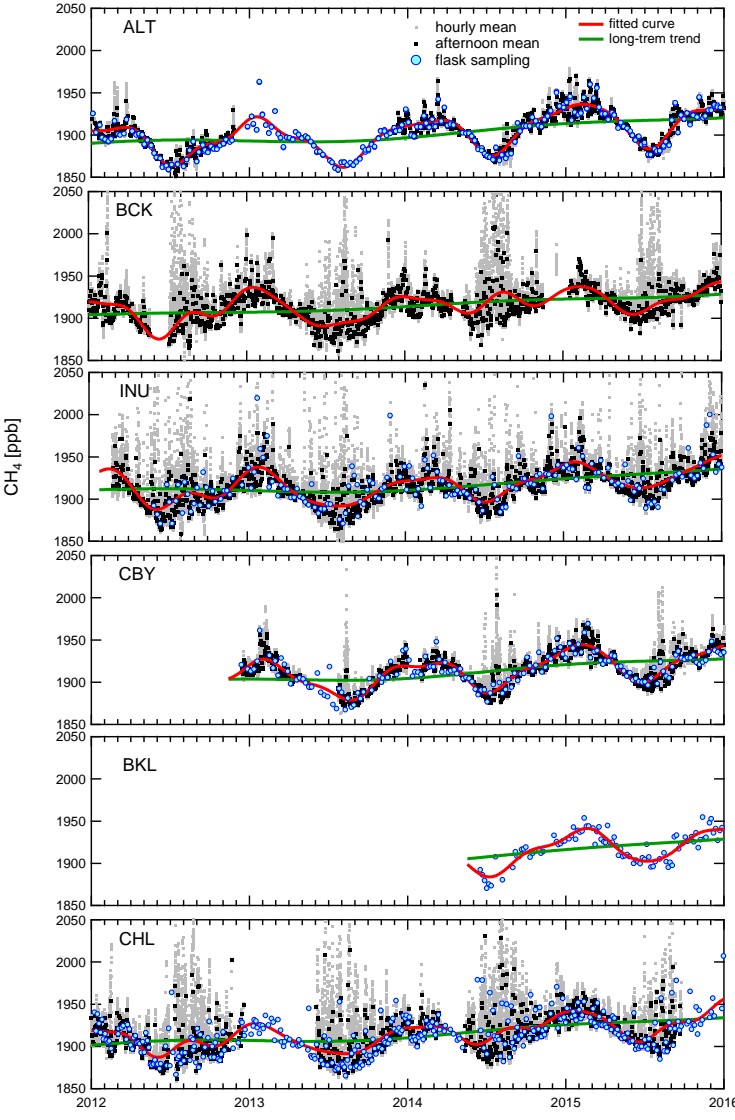

**Figure 2: Time-series of atmospheric CH$_4$ concentrations at Canadian Arctic sites. The observed values are the hourly means (grey dot), the afternoon means (black dot, 12:00−16:00 local time) from the continuous measurements and the ones from flask sampling (circle in light blue). BCK has only continuous measurements. At BKL, flask air sampling is only available after being initiated in 2014. The red and green curves are fitted curves and long-term trends which are obtained by applying a fitting-curve method to the observed afternoon means.**





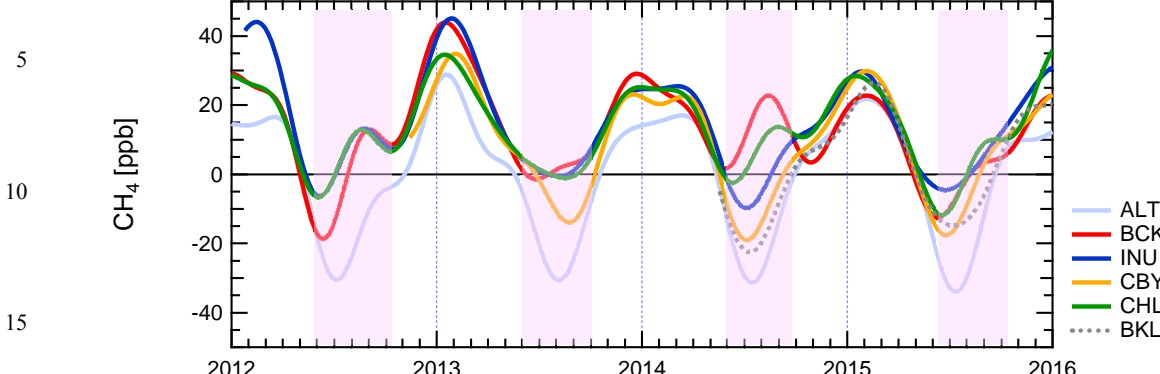

**Figure 3: Seasonal components in fitted curves of observed atmospheric CH$_4$ concentrations at Canadian Arctic Sites. Each fitted curve has subtracted the long-term trend component of Alert. Summer months (June–September) are highlighted by light pink shaded background.**





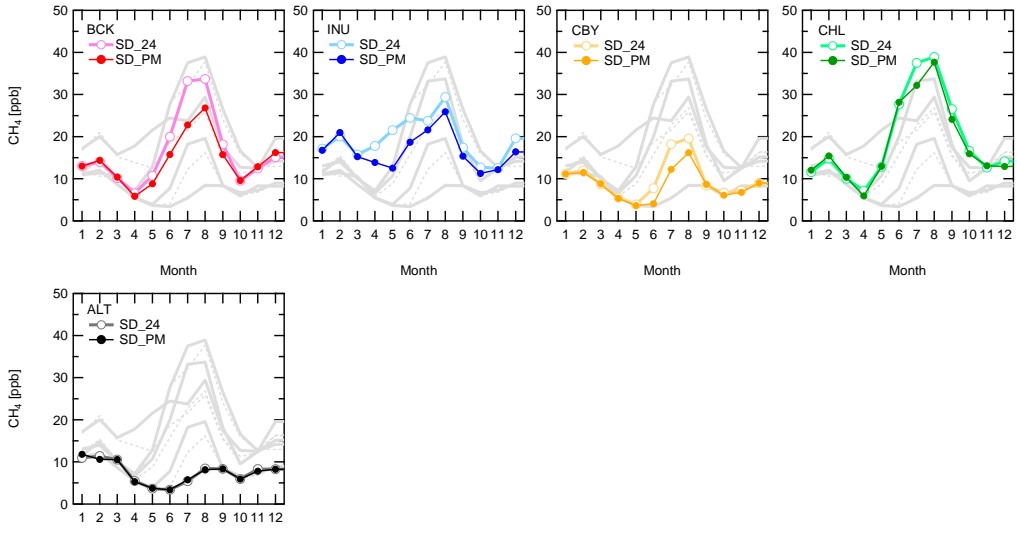

**Figure 4: Mean seasonal cycles of monthly standard deviation (SD) of observed CH₄ concentrations, SD_24 of all 24 hourly data (closed circles) and SD_PM of afternoon data (12:00−16:00 local time, open circles) to the fitted curves respectively.   For BCK, 2014 data have been excluded from the analysis, because of high variability due to massive large forest fires around the site.**





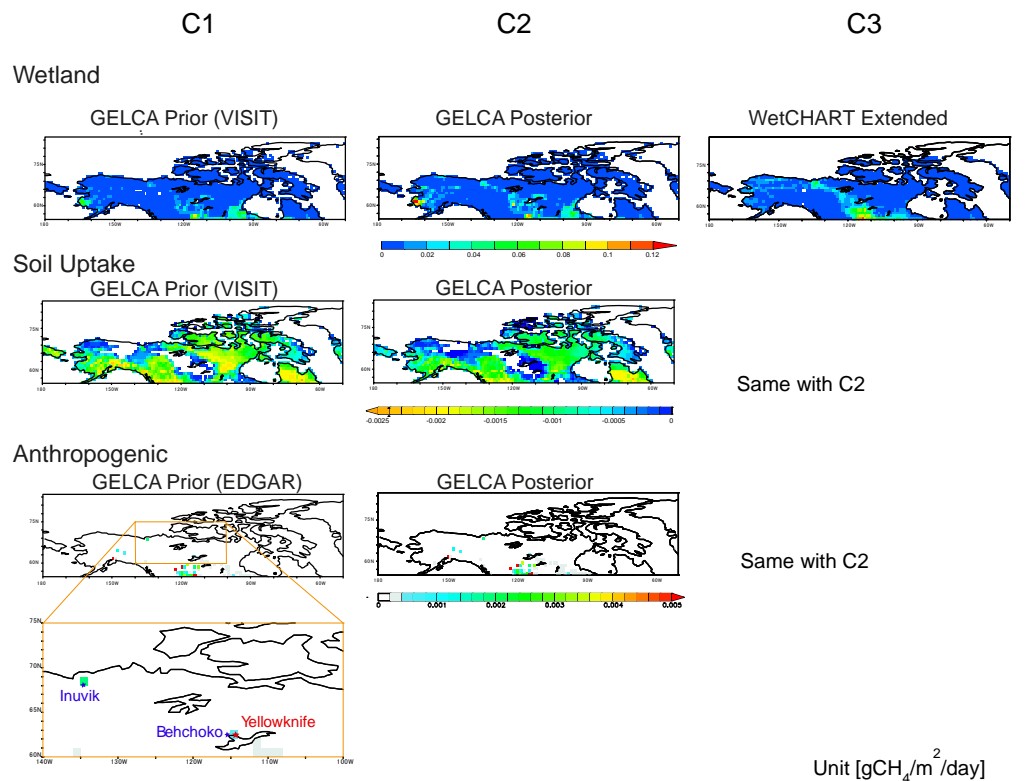

5   **Figure 5: Spatial distributions of summertime prior CH₄ fluxes of wetland emission, soil uptake and anthropogenic emissions for the three cases of prior fluxes, C1, C2 and C3, which are listed in table 2. Bottom left panel is a zoomed anthropogenic emission distribution in Northwest Territories. The locations of two sites, Behchoko (BCK) and Inuvik (INU) and the capital city, Yellowknife, are also plotted.**





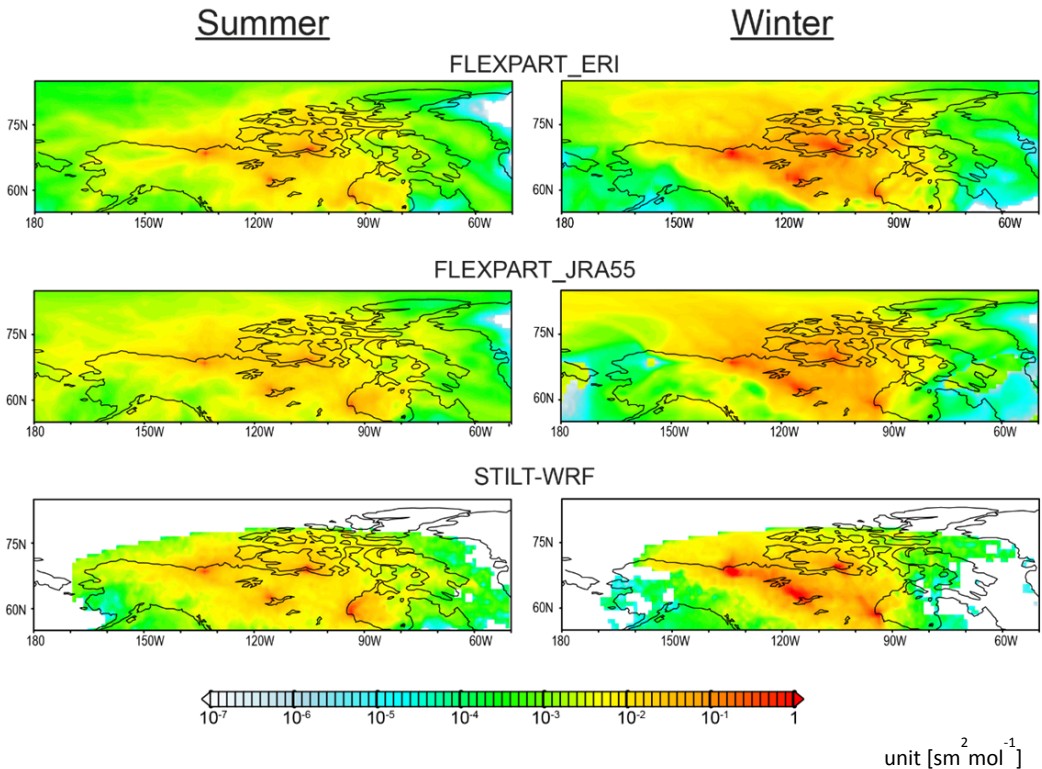

**Figure 6: Seasonal mean footprints of all sites by three models, shown for summer, July─August 2013) and winter (January─February 2013).**





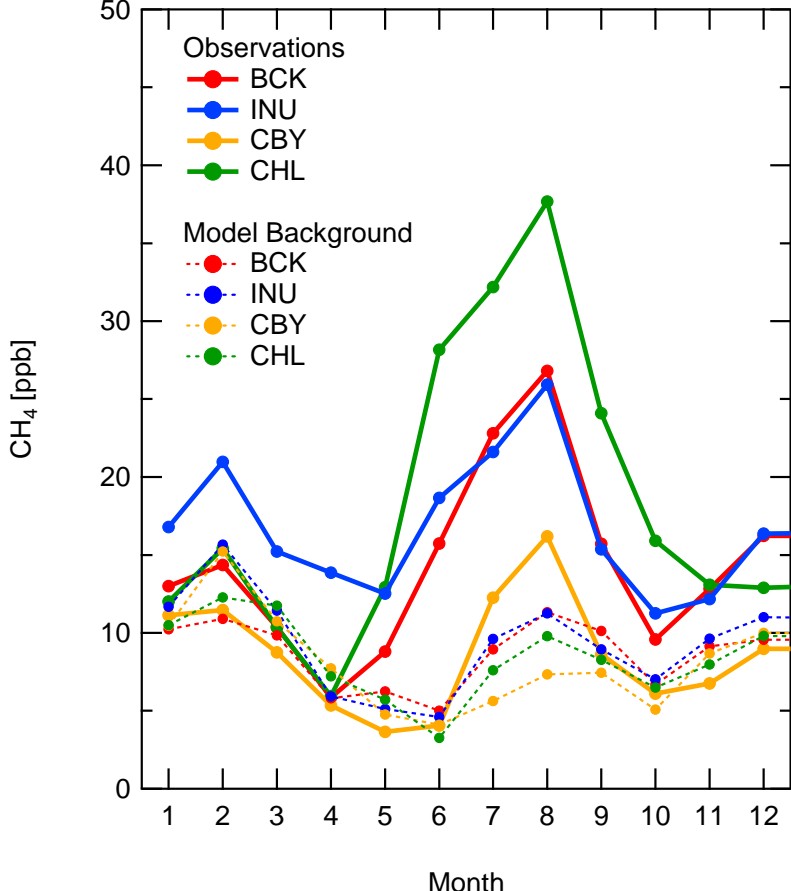

**Figure 7:** Four-year (2012−2015) mean monthly SD of modelled background CH$_4$ concentrations and SD of observed CH4 concentrations (afternoon data only, SD_PM). The background CH$_4$ concentrations are NIES-TM modelled concentrations weighted by the endpoints of 5-day back trajectory.





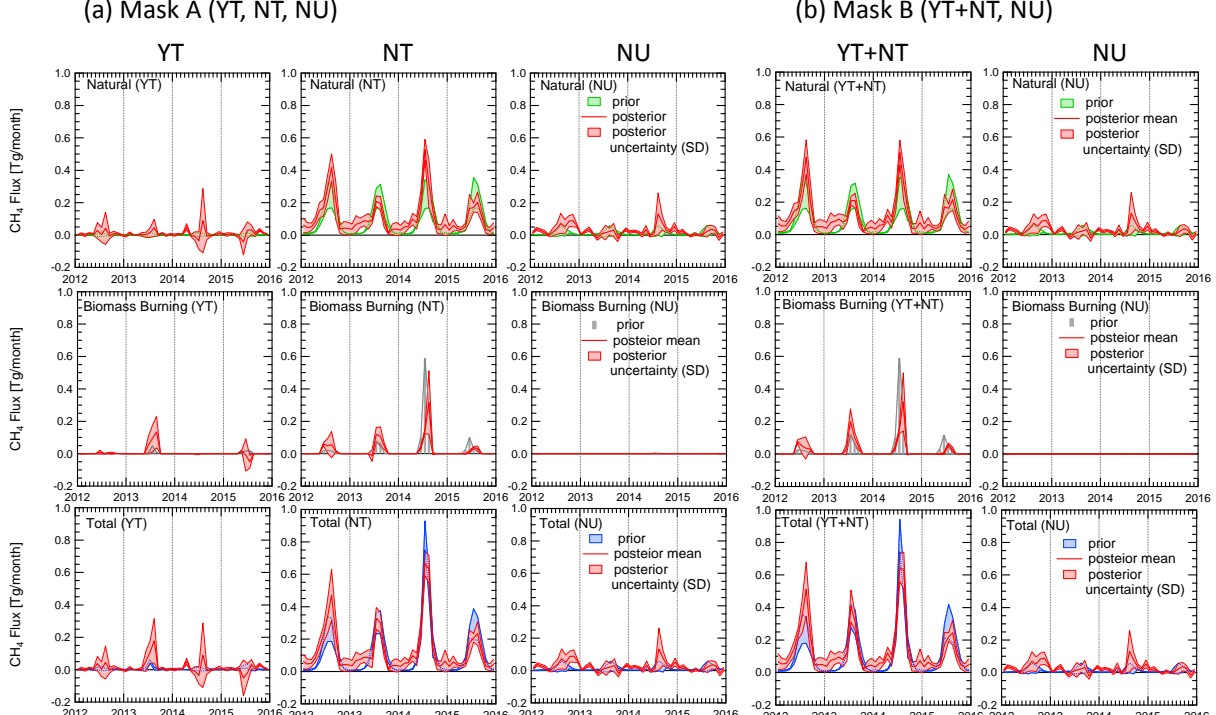

**Figure 8:** Monthly posterior mean fluxes with (a) sub-region Mask A (YT, NT, NU) and (b) Mask B (YT+NT, NU). Posterior mean flux is an average of nine experiments with 3 models (FLEXPART_EI, FLEXPART_JRA55 and WRF-STILT) and 3 prior emission cases (C1, C2 and C3). The posterior SD is shown by the red shaded area. Prior fluxes for natural include wetland flux, soil uptake and anthropogenic emissions. Biomass burning prior fluxes are from GFAS. The (non-red) shaded areas for natural and total prior fluxes indicate the range of prior fluxes.





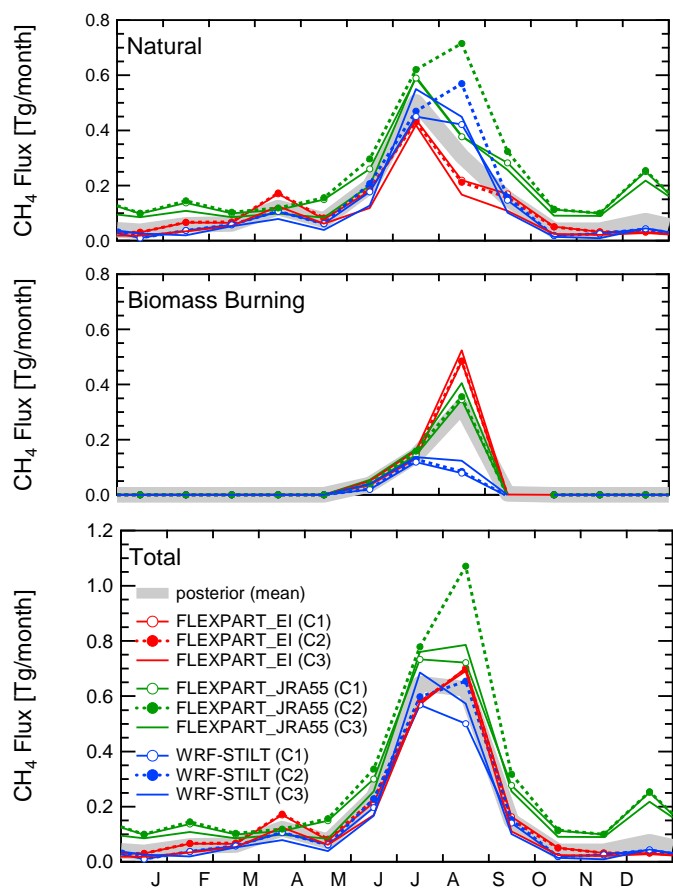

**Figure 9: Examples of monthly posterior fluxes by 9 inversion experiments of 3 different models (FLEXPART_EI, FLEXPART_JRA55 and WRF-STILT) with 3 prior emission cases (C1, C2 and C3). The posterior fluxes are plotted for sub-region YT+NT in Mask B. The posterior flux means over all nine experiments with Mask B are also plotted.**





(a) Mean annual Fluxes

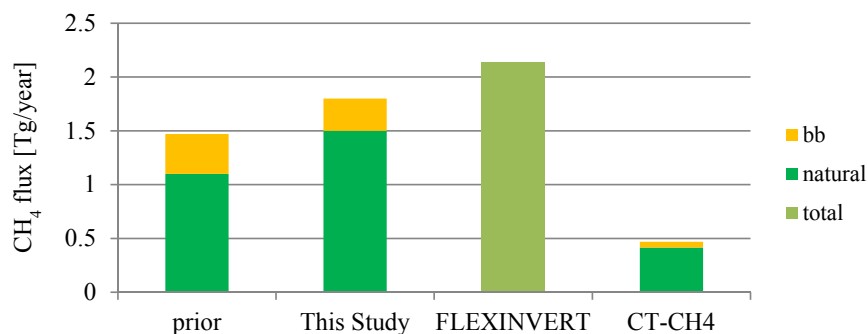

(b) Mean monthly Fluxes

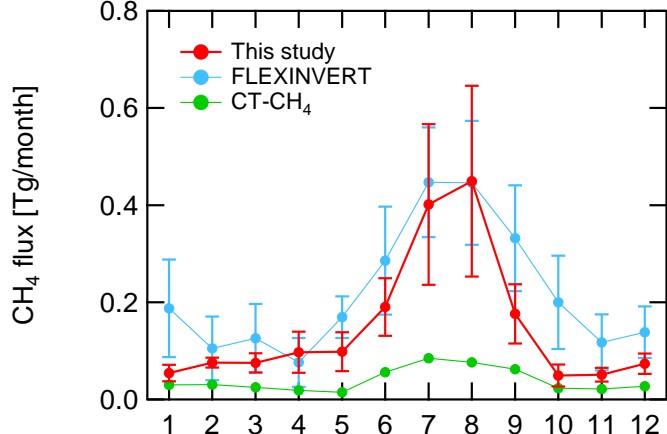

**Figure 10: mean prior and posterior (a) annual and (b) monthly fluxes for the Canadian Arctic. FLEXINVERT (Thompson et al., 2017) and CarbonTracker-CH₄ (CT-CH₄, Bruhwiler et al. (2014)) are plotted for comparison.**
10 **FLEXINVERT and CT-CH₄ fluxes are their last 5-year means, that is, 2009−2013 and 2006−2010 respectively. "natural" in CT-CH₄ are combined with the fluxes estimated as "anthropogenic" and "agriculture". The bars in monthly fluxes are SD of multi-year mean monthly fluxes.**



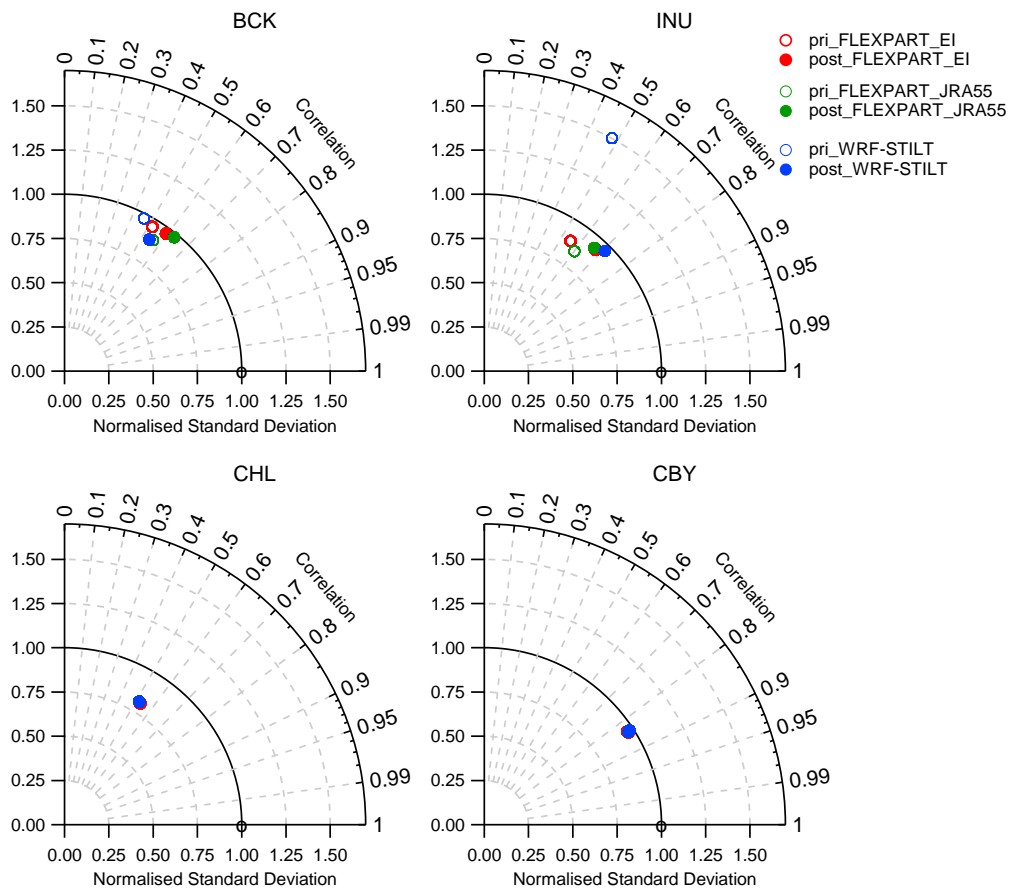

**Figure 11:** Taylor diagrams for the comparison between the prior (open circles) and posterior (closed circles) concentrations by three models: FLEXPART_EI (red), FLEXPART_JRA55 (green), and WRF-STILT (blue), with Mask B and prior flux case C3. The radius is the normalised standard deviation (NSD) of modelled concentrations against observations. The angle is the correlation coefficient. The values are the means with all observations and modelled concentrations per each simulation for each site.





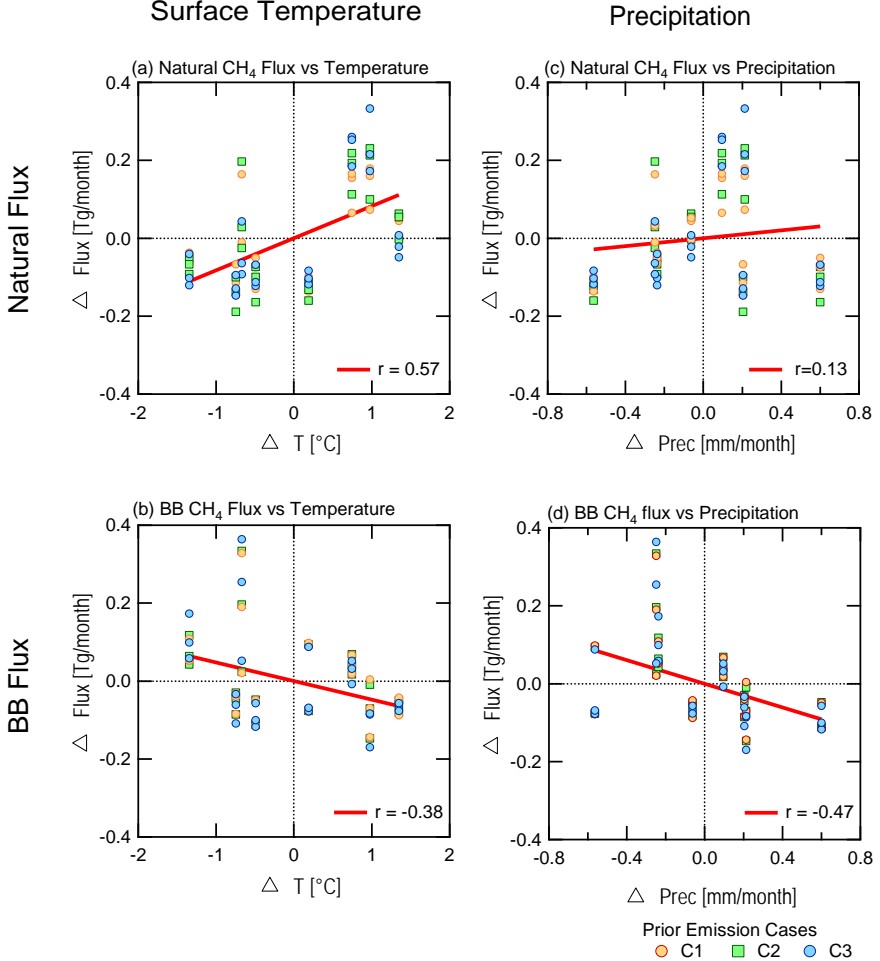

**Figure 12: CH₄ flux anomalies vs surface temperature and precipitation anomalies for summer (July and August). The CH₄ fluxes are July and August posterior fluxes for the Canadian Arctic from 9 inversion experiments with Mask C. Regional climate parameter anomalies in NT are monthly deviations from the four-year (2012─2015) means.**

