# Peer review of "Analysis of atmospheric CH4 in Canadian Arctic and estimation of the regional CH4 fluxes"

_Atmospheric Chemistry and Physics, 2018_

## Referee Comment (RC1) · Anonymous Referee #2 · 16 Nov 2018

Please see the attached pdf file.

Please also note the supplement to this comment:
https://www.atmos-chem-phys-discuss.net/acp-2018-907/acp-2018-907-RC1-supplement.pdf
* * *

---

## Referee Comment (RC2) · Anonymous Referee #1 · 24 Dec 2018

The authors have put together a really interesting paper on methane emissions from the Canadian Arctic. I think the datasets used in this study are really exciting data to work with, and I think recent monitoring efforts at Environment and Climate Change Canada could help move forward the field of high-latitude greenhouse gas fluxes. I have a few suggestions related to the paper. Most of these suggestions relate to better motivating the introduction and giving more prominence to the most important scientific results.

Overarching suggestions:

- In the intro, you explain that some of the uncertainty in greenhouse gas budgets could be due to different inverse modeling methodology. However, you use one flavor of Bayesian inverse modeling in the paper (albeit with different atmospheric models

and prior estimates). I think you could strengthen the intro by framing this discussion around uncertainties in transport and uncertainties due to the prior – topics that you actually explore in depth in the paper.

- I would be careful with the references throughout the text. In some cases, the references feel incomplete (particularly in the introduction), or you cite a reference that either did not focus primarily on that particular topic or was not the first to develop the concept.

- Most of the text in the results and discussion is dedicated to discussing more technical or methodological issues related to atmospheric transport, the inverse modeling setup, etc. I think the most interesting scientific conclusions of this paper are buried in Sect. 4.6.3 at the end of the results and discussion section. I would consider de-emphasizing some of the more methodological elements of the discussion and move the bigger science questions to a more prominent place. For example, you could pose the most important science questions at the end of the intro; that would give the reader an idea of what to expect. You could also move Sect. 4.6.3 to the beginning of the results and discussion and lengthen that section. You could also move the more methodological components of the discussion to the end and shorten that text.

Abstract:

- What provinces/territories or latitudes/longitudes do you define as the "Canadian Arctic"? That definition would help put the budget estimate in context.

- Abstract and throughout: The authors use the word "the" too often throughout the text. Some sentences would be smoother with fewer articles. For example, in line 9, "the regional CH4 flux" could be changed to "regional CH4 fluxes", and in line 10, "the recent observations" could be shortened to "recent observations." There are similar examples in most paragraphs of the manuscript.

Introduction:

- Pg. 2, lines 1-2: This sentence feels out of place. It does not summarize the content of the previous paragraph. Rather, it feels like the topic sentence of the paragraph starting in line 3.

- Pg. 2, lines 3-24: I would restructure these paragraphs. In the first two paragraphs, you state several times that methane fluxes are uncertain and only provide detail in the third paragraph. I would condense these three paragraphs into one and provide specific numbers sooner in the text.

- Pg. 2, line 25: What do you mean here by "the fluxes"? Are you referring to Arctic methane fluxes or greenhouse gas fluxes more broadly? The studies cited in this paragraph are not all methane studies. - Pg. 2, line 26: 4Dvar, Kalman filter, and geostatistical studies are all Bayesian.

- Pg. 2, lines 30-31: Can you provide references for this statement? Also, can you be more specific about how these differences have affected inverse modeling results in the past? What implications might those differences have for your study (i.e., for estimating Arctic methane)?

- Overall, the introduction includes a lot of broad, brush-stroke statements that some-times lack specifics, and it is not always clear how these statements concretely relate to the present study. I think you could strengthen the introduction two ways: (1) pro-vide more specific information to illustrate how uncertain or challenging these scientific questions are, and (2) Discuss why these uncertainties are particularly relevant to the present study or to understanding greenhouse gas fluxes from the Arctic.

- Pg. 3, lines 7-12: I think it would be stronger to frame this study around specific scientific questions instead of framing the study around presenting and analyzing ob-servations.

Measurements:

- Sect. 2.2: Some of this analysis might be a better fit for a results and discussion

section than a methods section. Furthermore, it seems like the main conclusions of this paper center around the inverse modeling results. Hence, I think some of this detail could go into a supplement.

Model description

- Sect. 3 title: Can you be more specific about which model you are referring to? The atmospheric transport model, the inverse model, or both?

- Pg. 7, line 13: I think the number of days required really depends upon the size of the domain and the geographic extent of the influence footprint.

- Pg. 7, line 24: Are you referring to a "model setting" or a "model setup"?

- Sect. 3.2: I think it would be helpful to have more descriptive flux model names than "C1", "C2", and "C3".

- Sect. 3.2: Somewhere in the text, it could be useful to include a sentence that explains why you chose these three particular prior models.

- Pg. 10, lines 22-23: How did you decide on these values of sigma?

- Pg. 10, line 24: What do you mean by "not strongly dependent"? Can you be more specific?

- Eq. 2: Lin et al. 2004 did not derive this equation and are not the first ones to use it. Instead, I would cite a textbook by Rodgers, Tarantola, or Enting.

- Pg. 11, line 7: The inverse model does not necessarily need to provide a perfect constraint on every region. Many modern inverse modeling studies estimate fluxes at model grid scale, even though the observations may not constrain each model grid box. If the observations do not provide a robust constraint at a particular location or time, the inverse modeling estimate will be guided by the prior estimate and the structure of the covariance matrix D_prior.

- Pg. 11, line 15: I think it would be useful to include one sentence explaining why you process the observations in this way.

Results and discussion

- Sect. 4.1: Why do you think the footprints are different, and is there one you think is better or more robust than another?

- Sect. 4.2: I don't think this information is essential to the paper – if you're looking to trim the text at all. Presumably, this information should also be reflected in the posterior uncertainties of the inverse model.

- Pg. 13, line 1: The word "significant" is often shorthand for "statistically significant." If you used a statistical test, I would clarify here with a p-value. If not, I would pick a different word than "significant" because that word may have specific meaning to many readers.

- Sect. 4.6.1: This result seems unsurprising to me. The inverse model includes several observing stations and more observations than unknowns. As a result, the prior and the covariance matrices do not need to do much "work" in the inverse model. I suspect that one would get similar estimates using a linear regression to estimate the scaling factors.

Summary:

- The summary feels like an extended abstract. It also repeats the description of some of the methodology. You might consider changing this section to a conclusions section and instead contextualize the results, discuss the possible implications of these results, and potentially make recommendations for future monitoring efforts in the North American or Canadian Arctic.

---

## Author Comment (AC1) · 5 Feb 2019

**Reply to Comments by Referee #1**

We thank the referee for constructive and helpful comments to improve our manuscript. We copy the comments in italic and red. In the responses, we also indicate the changes made in the manuscript (in blue font).

*Overarching suggestions*

*- In the intro, you explain that some of the uncertainty in greenhouse gas budgets could be due to different inverse modeling methodology. However, you use one flavor of Bayesian inverse modeling in the paper (albeit with different atmospheric models and prior estimates). I think you could strengthen the intro by framing this discussion around uncertainties in transport and uncertainties due to the prior – topics that you actually explore in depth in the paper.*

What we present in this paper is what we can see in the atmospheric $CH_4$ observations at our new Canadian Arctic measurement sites and how we can detect the regional $CH_4$ emissions with these new measurements given uncertainties in the inversion modelling framework used in this study. We have modified the introduction to state our objectives in this study more specifically. Detailed changes are explained in the following, as we answer to the specific comments.

*- I would be careful with the references throughout the text. In some cases, the references feel incomplete (particularly in the introduction), or you cite a reference that either did not focus primarily on that particular topic or was not the first to develop the concept.*

As we revised the manuscript, we replaced some references with more adequate ones and also added references. For example, we now refer methane emissions from permafrost to McGuire et al. (2009); Schuur et al.(2015); Thornton et al. (2016), instead of the IPCC report, Ciais et al (2013). Other changes are shown in our responses to the specific comments.

*- Most of the text in the results and discussion is dedicated to discussing more technical or methodological issues related to atmospheric transport, the inverse modeling setup, etc. I think the most interesting scientific conclusions of this paper are buried in Sect.4.6.3 at the end of the results and discussion section. I would consider de-emphasizing some of the more methodological elements of the discussion and move the bigger science questions to a more prominent place. For example, you could pose the most important science questions at the end of the intro; that would give the reader an idea of what to expect. You could also move Sect. 4.6.3 to the beginning of the results and discussion and lengthen that section. You could also move the more methodological components of the discussion to the end and shorten that text.*

This section [Relationship of fluxes with climate anomalies] is scientifically interesting regarding climate change, in this paper. Before we get to the point, we first need to understand the robustness and uncertainties of the estimated regional $CH_4$ fluxes. That is why we have conducted inversion experiments using multiple atmospheric models, prior fluxes, and sub-region masks. Furthermore, this paper aims to present the new observational data of atmospheric $CH_4$ in the Canadian Arctic and the inferred regional fluxes utilising the new atmospheric observations. We anticipate

expanding this study to quantify a trend in $CH_4$ emissions as well as inter-annual variations in the Canadian Arctic with longer observational records in the future.

As suggested, we moved Section 4.6.3 forward (not to the beginning, but to Section 4.5) in Results and Discussions, after Section 4.4 (Comparison with previous estimates), where we mention the interanual variability of estimated fluxes, comparing with the results from CARVE. We lengthen this section (now Section 4.5) by adding the discussion on influence of prior fluxes.   Note that the 3 different prior fluxes C1, C2 and C3 have been changed to the more descriptive names VIS, GEL and WetC following a specific comment. The previous Figure 12 is updated and shifted to Figure 11 to be consistent with the new modifications. These new names appear in the following:

**4.5  Relationship of fluxes with climate anomalies**

Inter-annual variations of estimated $CH_4$ fluxes are examined in relation to climate parameters here, specifically with surface air temperature and precipitation from NCEP reanalysis (Kalnay et al., 1996).  First, monthly mean values at the sub-regions as well as the 4-year mean (2012-2015) for each month are calculated, then the monthly anomalies are computed from the monthly mean values and the 4-year mean of the corresponding month.   The temperature and precipitation anomalies are aggregated to the respective regions, NT, YT and NU.  On the regional level, climate anomalies in NT and NU are quite similar, though YT is less similar to NT and NU.  YT is mainly covered by mountains with little wetland. Furthermore, NT has the largest wetland extent and most of the forest fire emissions in 2012-2015.  Thus, we look into the correlation in monthly anomalies of $CH_4$ fluxes with summer climate anomalies in NT.

In Fig. 11, the inter-annual variability of wetland $CH_4$ fluxes exhibits a moderate positive correlation with the surface temperature anomaly (r = 0.55) and only weakly correlated with precipitation anomalies (r = 0.11).  This indicates that the hotter summer weather condition stimulates the wetland $CH_4$ emission, and precipitation appears to have a less immediate or no direct impact on wetland conditions.  In prior cases VIS and GEL, natural $CH_4$ fluxes (wetland and other fluxes except biomass burning $CH_4$ flux) are multi-year mean monthly fluxes.  Therefore these prior fluxes have no year-to-year anomalies and no correlation with the meteorological anomalies.  Only in WetC, the prior with wetland $CH_4$ fluxes from WetCHARTs ensemble mean exhibits inter-annual variations, the correlations with temperature and precipitation anomalies are r = 0.26 and r = 0.90 respectively.  The posterior natural fluxes with WetC show slightly higher correlations (r=0.55 with temperature, r=0.16 with precipitation) than the mean correlation values.  But overall there is no clear dependency of posterior correlations on the inherent climate anomaly correlations in the prior fluxes.  This result indicates that the inter-annual variations in posterior wetland fluxes in this study are mainly determined by the observations, rather than by prior fluxes.

Inter-annual variations of estimated BB $CH_4$ fluxes show a negative correlation with precipitation (r = -0.41).  Also throughout the fire season (June-September), all estimated BB fluxes negatively correlate with precipitation while the prior BB fluxes appear to have no consistent correlations. The inversion results support that dry condition would enhance the forest fire.  The estimated BB fluxes show weakly negative correlation with surface temperature (r = -

0.23) on mid-summer average, but the monthly correlations are fluctuating from r = -0.40 to r = 0.47 over the fire season. Since the period is limited in this study (2012─2015), these statistical relationships are still not clear. Also, the relationship of $CH_4$ emissions with climate conditions could be complex and non-linear (with extreme fires events in some years). More data and analysis are required to characterise the dependence of $CH_4$ fluxes on climate in the Arctic.

*Abstract:*

*- What provinces/territories or latitudes/longitudes do you define as the "Canadian Arctic"? That definition would help put the budget estimate in context.*

We added latitude information (>60˚N, 60˚W─141˚W).  Since the Canadian province and territories might not be familiar to some readers, it would be better to be introduced later, not in abstract.

*- Abstract and throughout: The authors use the word "the" too often throughout the text. Some sentences would be smoother with fewer articles. For example, in line 9, "the regional CH4 flux" could be changed to "regional CH4 fluxes", and in line 10, "the recent observations" could be shortened to "recent observations." There are similar examples in most paragraphs of the manuscript.*

We tried to omit "the' if it is not necessary in Abstract and throughout the manuscript to improve the readability.

*Introduction:*

*- Pg. 2, lines 1-2: This sentence feels out of place. It does not summarize the content of the previous paragraph. Rather, it feels like the topic sentence of the paragraph starting in line 3.*

We have moved the sentence to the next paragraph as a topic sentence, which is shown in the response to the next comment.

*- Pg. 2, lines 3-24: I would restructure these paragraphs. In the first two paragraphs, you state several times that methane fluxes are uncertain and only provide detail in the third paragraph. I would condense these three paragraphs into one and provide specific numbers sooner in the text.*

Following your comment, we have re-structured these paragraphs. Now the original third paragraph is the top and the first and second ones follow with modifications.  Top sentence of the new first paragraph has been moved in from the immediate previous paragraph as answered to the previous comment:

[revised manuscript text omitted]

*- Pg. 2, line 26: 4Dvar, Kalman filter, and geostatistical studies are all Bayesian.*

Yes, all of them are variations of Bayesian inversion. We do not intend to explore the details of different inversion schemes (as we used only one type of Bayesian scheme). As seen in our response to the previous comment, we have removed these detailed technical terms from the revised text.

*- Pg. 2, lines 30-31: Can you provide references for this statement? Also, can you be more specific about how these differences have affected inverse modeling results in the past? What implications might those differences have for your study (i.e., for estimating Arctic methane)?*

This statement listed some of the possible sources of uncertainties in inversion results. We agree it is too general to be informative and it has been deleted in the revision.

*- Overall, the introduction includes a lot of broad, brush-stroke statements that sometimes lack specifics, and it is not always clear how these statements concretely relate to the present study. I think you could strengthen the introduction two ways: (1) provide more specific information to illustrate how uncertain or challenging these scientific questions are, and (2) Discuss why these uncertainties are particularly relevant to the present study or to understanding greenhouse gas fluxes from the Arctic.*

As we answer in the next comment, we have modified the last two paragraphs to state more specifically what are challenging for estimating the Arctic greenhouse gas fluxes and our monitoring effort, as narrowing down to what we present in this study.

*- Pg. 3, lines 7-12: I think it would be stronger to frame this study around specific scientific questions instead of framing the study around presenting and analyzing observations.*

The last paragraph in the Introduction has been revised to focus more on the scientific questions as follows:

This is the first study to analyse the atmospheric $CH_4$ mixing ratios from the above new ECCC observation sites in the Canadian Arctic region. In this study, we address three key questions: (1) what can the new measurements see from local and regional sources? (2) what are the estimated $CH_4$ fluxes in the Canadian Arctic from inverse modelling using these new measurements? and (3) are there any relationship for the Canadian Arctic $CH_4$ fluxes with climate/environmental variations? This paper is structured as follows: In Section 2, the description of the measurement stations as well as the observational data analyses from daily to inter-annual time scales are given. Section 3 describes the inversion model framework, and Section 4 presents flux estimates and discusses the flux uncertainties and relationship to climate anomalies.

*Measurements:*

*- Sect. 2.2: Some of this analysis might be a better fit for a results and discussion section than a methods section. Furthermore, it seems like the main conclusions of this paper center around the inverse modeling results. Hence, I think some of this detail could go into a supplement.*

Since this is the first paper to analyse these new measurement data, we thought it is better to present the data analyses in the main text in a (mostly) self-contained section (for readers with more interest on the data characteristics than inversion results). We do show some analysis of the observed data with the model results in Section 4.2 [Signals in the observations (relative to the background) ] where it is more appropriate.

*Model description*

*- Sect. 3 title: Can you be more specific about which model you are referring to? The atmospheric transport model, the inverse model, or both?*

The title is changed to 'Regional inversion model description'. We hope this is clearer as the atmospheric transport model (Section 3.1) is one component of the regional inverse model. We used different combinations of atmospheric transport model and atmospheric forcing data to help estimate the transport uncertainties in the inversion results.

*- Pg. 7, line 13: I think the number of days required really depends upon the size of the domain and the geographic extent of the influence footprint.*

Yes. Particle traveling time and size of domain of the interest are related. Bigger space/domain need more time (for the air particles to travel over). As seen in Figure 6, the 5-day footprint cover the Canadian Arctic, the domain of our interest and the location of the particles after 5 days are mostly outside the Canadian Arctic. Furthermore most synoptic-scale variations in the $CH_4$ mixing ratios at measurement sites are sufficiently explained by footprints within 2-5 days after particles are released.

*- Pg. 7, line 24: Are you referring to a "model setting" or a "model setup"?*

Since we describe the combinations of transport models and meteorological data that we used to obtain footprints, "model setup" would be more suitable than "model setting". We have replaced "model setting" with "model setup" here and other occurrences.

*- Sect. 3.2: I think it would be helpful to have more descriptive flux model names than "C1", "C2", and "C3".*

Each flux scenario consists of some different fluxes, but some common fluxes. The distinctive features are wetland $CH_4$ fluxes. To reflect the differences in Wetland $CH_4$ emission, we modified C1, C2 and C3 to VIS, GEL and WetC, respectively, as follows:

From C1 to VIS     : VISIT CH4 wetland model,

From C2 to GEL    : optimised fluxes by GELCA-CH$_4$ inversion

From C3 to WetC   : WetCHARTs

*- Sect. 3.2: Somewhere in the text, it could be useful to include a sentence that explains why you chose these three particular prior models.*

We chose the first two cases of prior fluxes since the background atmospheric $CH_4$ fields is calculated by a global model NIES-TM with GELCA-$CH_4$ optimised fluxes. However, these fluxes are averaged to be cyclo-stationary (climatological). WetCHARTs provides inter-annually varying wetland $CH_4$ fluxes for the study period. $CH_4$ emissions of WetCHARTs are driven by environmental parameters, including satellite-based wetland extent.

The first paragraph of Section.3.2 has been revised as follows:

Three cases of prior emissions, VIS, GEL and WetC, were used as listed in Table 2. Since the global background atmospheric $CH_4$ field is calculated with GELCA-$CH_4$ inversion posterior fluxes, we chose the prior (VIS) and posterior (GEL) fluxes from the GELCA global inversion as two cases of prior fluxes in our regional inversion, respectively. Note that the continuous $CH_4$ mixing ratio data from the new Canadian Arctic sites were not used in the GELCA-$CH_4$ inversion. In this study, the mean wetland fluxes for the last 5 years of the GELCA global model were used, the prior forest fire $CH_4$ fluxes are detailed in Section 3.2.2. The third prior case (WetC) is the same as GEL, but with wetland $CH_4$ fluxes from WetCHARTs (a recent global wetland methane emission model ensemble for use in atmospheric chemical transport models). WetCHARTs provide inter-annually varying monthly wetland $CH_4$ fluxes for this study period. The details of prior fluxes are described in the following sections

*- Pg. 10, lines 22-23: How did you decide on these values of sigma?*

The sigma value for prior uncertainty, $\sigma_{prior}$=0.30, is from the uncertainty in the $CH_4$ emission used in Zhao et al. (2009). The prior model-mismatch, $\sigma_e$=0.33, is comparable to those used in previous regional inversion studies (e.g., Gerbig et al., 2013; Lin et al., 2004, Zhao et al., 2009), which considered different error components such as wind field, aggregation, and background $CH_4$ mixing ratio. We examined the sensitivity to these sigma values as explained in the next comment/response.

We have added the references for these sigma values in the revised manuscript.

*- Pg. 10, line 24: What do you mean by "not strongly dependent"? Can you be more specific?*

We have added more details to clarify lines 23-24:

We examined the inversion's sensitivity to these uncertainties by doubling their values. The posterior fluxes changed by less than 5% for all sub-regions (and the different sub-region masks). The results showed the optimised fluxes are not strongly dependent on these prescribed uncertainties.

*- Eq. 2: Lin et al. 2004 did not derive this equation and are not the first ones to use it. Instead, I would cite a textbook by Rodgers, Tarantola, or Enting.*

It is true that Lin et al. (2004) did not introduce this equation. We have added three textbooks as references for the Bayesian inversion approach at the beginning of the section: Tarantola (1987), Rodgers (2000) and Enting (2002). They collectively cover the basic concept and equations of Bayesian inversion and its applications of atmospheric greenhouse gases. We have removed the reference Lin et al.(2004).

*- Pg. 11, line 7: The inverse model does not necessarily need to provide a perfect constraint on every region. Many modern inverse modeling studies estimate fluxes at model grid scale, even though the observations may not constrain each model grid box. If the observations do not provide a robust constraint at a particular location or time, the inverse modeling estimate will be guided by the prior estimate and the structure of the covariance matrix D_prior.*

Yes, we do agree that it is possible to estimate fluxes for sub-regions not well constrained by observations, by using the additional information in the '*prior estimate and the structure of the covariance matrix D_prior*'. Given all the constraints, the estimated fluxes are still susceptible to (unaccounted for) errors such as model transport biases, non-Gaussian error distributions and other problems. In this study, we found that the weak flux region (Yukon) could not be robustly constrained (fluctuating from positive to negative fluxes). Hence, we explored different sub-region masks to test the robustness of the inversion results. We have clarified the statement with an addition as follows:

From:
 The inversion results in the next section will show YT could not be reliably constrained as a separate sub-region.

To:
 The inversion results in the next section will show YT could not be reliably constrained as a separate sub-region (model uncertainties made the estimated fluxes in YT fluctuate from positive to negative).

*- Pg. 11, line 15: I think it would be useful to include one sentence explaining why you process the observations in this way.*

We have added an explanation of the afternoon mean values:

First, the afternoon mean values are calculated by averaging the hourly data over 4 hours from 12:00 to 16:00 local time so that the observations we use in this study are more regionally representative assuming mid-day is in well-mixed planetary boundary layer. Second, the modelled background mixing ratio, which were described earlier, are subtracted from the afternoon means.

*Results and discussion*
*- Sect. 4.1: Why do you think the footprints are different, and is there one you think is better or more robust than another?*

The footprints are dependent on the meteorological fields, parametrized dispersions, etc. Thus the 'quality' or 'goodness' of the footprints could vary with time and place (as a function of the 'quality' of the meteorological forcings and dispersion model parametrizations, etc.). In the Canadian Arctic (particularly in summer), we found that for spatially distributed and slow varying fluxes like wetland CH4, the results are less sensitivity to the transport model difference, but fast varying and point like sources like biomass burning are quite sensitive to the model transport differences. Since it is difficult to separate the errors in transport and errors in emissions when comparing modelled and observed mixing ratios, we are mainly using the different transport models to provide an estimate of transport uncertainties in the inversion results.

*- Sect. 4.2: I don't think this information is essential to the paper – if you're looking to trim the text at all. Presumably, this information should also be reflected in the posterior uncertainties of the inverse model.*
We think this section could be informative for the general reader (not all working on inversion modelling). It provides an evaluation of the amount of the regional $CH_4$ signals in the model compared to the 'background' and an explanation on the relatively larger percentage error in the winter inversion results compared to the summer. Thus, Section 4.2 is potentially useful, we have decided to keep this section.

*- Pg. 13, line 1: The word "significant" is often shorthand for "statistically significant." If you used a statistical test, I would clarify here with a p-value. If not, I would pick a different word than "significant" because that word may have specific meaning to many readers.*
We have changed "significant" to "noticeable'.

*- Sect. 4.6.1: This result seems unsurprising to me. The inverse model includes several observing stations and more observations than unknowns. As a result, the prior and the covariance matrices do not need to do much "work" in the inverse model. I suspect that one would get similar estimates using a linear regression to estimate the scaling factors.*
It seems that our inversion results are not sensitivity to +/- 50% change in prior emission due to having sufficient regional signals compared to the background conditions. The summertime regional signals in atmospheric $CH_4$ is strong enough to infer regional fluxes, as we show in Section 4.2 [Signals in the observations (relative to background)]. In this section, we are trying to demonstrate that the results appear robust. We do agree that a 'linear regression estimate to the scaling factors' should work in this case. Given the strong observational constraints, the prior constraint term is likely not important.

*Summary:*
*- The summary feels like an extended abstract. It also repeats the description of some of the methodology. You might consider changing this section to a conclusions section and instead contextualize the results, discuss the possible implications of these results, and potentially make recommendations for future monitoring efforts in the North American or Canadian Arctic.*

We have changed the section title from summary to conclusions.  We have also shortened the texts of the results, but stated more on a future direction and the implication of the study results as follows:

**5 Conclusions**

The Canadian Arctic region is one of the potential enhanced $CH_4$ source regions related to the ongoing global warming (AMAP, 2015), and earth system models differ in their prediction how the carbon loss there will be split up between $CO_2$ and $CH_4$ emissions. Even current bottom-up and top-down estimates of the $CH_4$ flux in the region vary widely. This study:

1) analysed the measurements of atmospheric CH4 mixing ratios that include 5 sites established in the Canadian Arctic by ECCC, to characterise the observed variations and examine the detectability of regional fluxes.

And,

2) estimated the regional fluxes for 4 years (2012─2015) with the continuous observational data of atmospheric $CH_4$, and also the relationship of the estimated fluxes with the climate anomalies.

The observational data analysis reveals large synoptic summertime signals in the atmospheric $CH_4$, indicating strong regional fluxes (most likely wetland and biomass burning $CH_4$ emissions) around Behchoko and Inuvik in Northwest Territory, the western Canadian Arctic.  The local signals of atmospheric $CH_4$ also allow inverse models to optimise biomass burning $CH_4$ flux (emissions due to forest fire), separately from the remaining/natural $CH_4$ fluxes (including wetland, soil sink and anthropogenic, but mostly due to wetland $CH_4$ emissions).

The estimated mean total $CH_4$ annual flux for the Canadian Arctic is $1.8 \pm 0.6$ TgCH$_4$ yr$^{-1}$ (wetland flux is $1.5 \pm 0.5$ TgCH$_4$ yr$^{-1}$, biomass burning flux $0.3 \pm 0.1$ TgCH$_4$ yr$^{-1}$). The mean total flux in this study is comparable to another regional flux inversion result of 2.14 TgCH$_4$ yr$^{-1}$ by Thompson et al. (2017), but much higher than the global inversion result of 0.5 TgCH$_4$ yr$^{-1}$ by CarbonTracker-CH$_4$ (Bruhwiler et al., 2014).  The strong regional $CH_4$ signals at INU and BCK appear to yield flux estimates in this study with narrower high summer emission period and lower wintertime wetland emission compared with the estimates by Thompson et al. (2017).

Clear inter-annual variability is found in all the estimated summertime natural $CH_4$ fluxes for the Canadian Arctic, mostly due to Northwest Territories. These summertime flux variations are positively correlated with the surface temperature anomaly (r = 0.55). This result indicates that the hotter summer weather condition stimulates the wetland $CH_4$ emission.

With longer data records and more analysis in the Arctic, inversion $CH_4$ flux estimates could yield more details on $CH_4$ emission strength and seasonal cycle (onset and termination of wetland emissions), and dependence of wetland fluxes on climate conditions. More knowledge on the flux and climate relationship could help evaluate and improve bottom-up wetland $CH_4$ flux models.

Next, we will perform a similar study for the $CO_2$ measurements from these sites to estimate the Canadian Arctic $CO_2$ fluxes. Estimation of $CO_2$ and $CH_4$ fluxes and monitoring how these fluxes change in the future will improve our understanding on the response of the Arctic carbon cycle to climate change, and also yield long-term trends in $CO_2$ and $CH_4$ emissions in the Canadian Arctic.

---

## Author Comment (AC2) · 5 Feb 2019

**Reply to Comments by Referee #2**

We thank the referee for constructive and helpful comments to improve our manuscript with more clarifications. We address the comments (italic and red). In the responses, we also indicate the changes made in the manuscript (in blue font).

*General comments*

*This manuscript presents atmospheric observations of CH4 from 5 new in-situ measurements sites in the Canadian Arctic and uses these (plus one other site) in atmospheric inversions to determine the land-atmosphere flux of CH4. The authors find notable inter-annual variability in the natural (wetland) flux, which may be related to variations in surface temperature. Although the study is interesting and fairly well presented, further explanations and clarifications for some of the methods are needed before being published. In addition, minor technical corrections for English language usage are required.*

We report the modifications and additional explanations that we made in the manuscript.

**Specific comments**

P1, L14: I suggest specifying the number, instead of "multiple". Also how is "inversion modelling system" defined, by the inversion algorithm or transport model used? In this study 2 different transport models were used with 3 different meteorological datasets, so I suggest the authors state this instead.

As suggested, we have changed the sentence more specifically:

From:

Multiple regional Bayesian inversion modelling systems are applied..

To:

Three regional Bayesian inversion modelling systems with two Lagrangian Particle Dispersion Models and three meteorological datasets are applied…

*P1, L30: I suggest the authors state what the carbon is vulnerable to, i.e., conversion to CH4 and CO2 which can be emitted to the atmosphere*

After "vulnerable", we have added "to conversion to $CH_4$ and $CO_2$ which can be emitted to the atmosphere".

*P2, L5: Please specify the magnitude of what, presumably CH4 emission but this should be stated*

Yes, it is the magnitude of $CH_4$ emission. We have changed to:

"… show large discrepancies in the spatial distribution of wetland $CH_4$ source, as well as its magnitude"

*P3, L7 (and throughout): It's actually CH4 volume mixing ratio that is reported, and not concentration, so I suggest changing "concentration" to "mixing ratio" throughout.*

As the referee noted, gaseous concentration is frequently referred to as 'volume mixing ratio'. We have changed from "concentration" to "mixing ratio", throughout the manuscript.

*P5, L31: By "SD of the observed time series to their fitted curves" do the authors mean the SD of the residuals, i.e., after subtracting the fitted curves? This is not clear.*

Yes, we mean SD (standard deviation) of the residual of the observations from a fitted curve. That is also referred residual standard deviation. For clarification, we have added "residual" into the sentence:

 "monthly Standard Deviation (SD) of the residual of observed time series..."

*P5, L34 to P6, L2: This needs some explanation why the difference between SD_PM and SD_24 gives an indication of whether the daily variability is due to local scale changes in emissions or seasonally changing atmospheric transport. I guess the authors mean that SD_24, which includes also night-time data, is more sensitive to local emissions than SD_PM, but an explanation should be provided.*

We use the difference between SD_PM and SD_24 as an indication of local emission. SD_24 includes nighttime data which is more sensitive to local emissions than SD_PM. For clarification, we have added the following sentences:

The nighttime planetary boundary layer (PBL) is usually shallow, while the daytime boundary layer is usually deeper and well mixed. If there are local $CH_4$ sources, the emission is mixed into a shallow PBL at night (yielding higher mixing ratio) and deeper PBL during the day (yielding lower mixing ratio). The resultant diurnal variations in the $CH_4$ mixing ratios are evident as larger $CH_4$ SD_24 compared to SD_PM. In the absence of local sources, SD_24 is comparable to SD_PM.

*P6, L6: "rectified" is not the right term here (the rectifier effect is a specific term given to the co-variation of flux and planetary boundary layer height, particularly for CO2, which doesn't apply here). Instead use "amplified".*

As the referee suggested, "amplified" is more suitable than "rectified". We have revised the text accordingly.

P10, L7: The authors should change this sentence to either "Our Bayesian inversion optimizes…" or "The Bayesian inversion used here optimizes…" to make it clear that the approach used here is not the only approach.

We have changed:

From: "The Bayesian inversion optimises…"

To : "The Bayesian inversion used here optimises…"

*P10, L16: The authors state that the matrix $K$ is the product of $M$ (the footprints) and $x$ (the surface fluxes) and is a Jacobian matrix of flux sensitivities. The elements of $K$ must be in mass mixing ratio units (i.e. the same units as $y$), so by definition this is not a Jacobian matrix (but $M$ is a Jacobian). Also, the dimensions of $M$ and $x$ should be given.*
As the referee noted, our usage of 'Jacobian matrix of flux sensitivities' is not correct. To avoid confusion, we have removed the term-"Jacobian matrix" from the sentence and revised with the addition of the dimensions of $M$ and $x$. The text has been changed:

From:
$K$ is the matrix of contributions from R sub-regions. $K$ is a Jacobian matrix of flux sensitivity, a product of two matrices, $M$ and $x$. $M$ is the modelled transport (or footprints in this study), and $x$ is the spatial distribution of the surface fluxes.

To :
$K$ (N×R) is the matrix of contributions on the observations (N) from all the fluxes (R) of sub-regions. $K$ is a product of two matrices, $M$(N × LL) and $x$ (LL×R), $M$ is the modelled transport (or footprints in this study), and $x$ is the spatial distribution of the surface fluxes. LL (=LAT×LON) is the dimension of our domain (1°x1° in latitudes (LAT) by longitudes (LON)).

*P10, L22: The units of the observation uncertainty should be specified, presumably this is ppb. Also, an explanation should be given of how the value of 0.33 was derived, especially as this seems rather small. Furthermore, an estimate of the appropriateness of the uncertainty estimates should be given, e.g. the value of the reduced-chi-square statistic.*
The 33% (0.33) prior model-data mismatch is comparable to other regional inversion studies (e.g. Gerbig et al. (2003), Zhao et al. (2009)). Zhao et al. (2009) included uncertainties from LPDM dispersion, wind field, aggregation and background mixing ratio to estimate prior model-data mismatch uncertainty. However such estimate has many assumptions that are difficult to evaluate. In this study, we tested the sensitivity of the inversion results to this setting by using 33% and 66%, as the model-data mismatch errors. The posterior fluxes changed by less than 5% for all sub-regions (and the different sub-region masks), indicating that the flux estimates were not highly sensitive to the prior error specification.

In response to a similar comment from referee 1, we have added more details (in blue below) to clarify page 10, lines 23-24:

'We examined the inversion's sensitivity to these uncertainties by doubling their values. The posterior fluxes changed by less than 5% for all sub-regions (and the different sub-region masks). The results showed the optimised fluxes are not strongly dependent on these prescribed uncertainties.'

In statistical error analysis, the reduced chi-square test qualitatively measures the goodness of fit of the model to the observations (Hughes and Hase, 2010, Drosg (2009)). In the limit of infinite number of data points and the data are independent and normally distributed, the value of reduced chi-square should be 1. Following Drosg (2009), the reduced chi-square is given by:

$$reduced\ \chi^2 = \frac{1}{N} \sum_k \sum_j {\chi_{jk}}^2$$

Where chi-square is:

$$\chi_{jk}^2 = \frac{\sum_i (model_{ijk} - obs_{ijk})^2}{\sigma_{jk}^2}$$

$$obs = observation - model_{background}$$

$$\sigma_{jk} = standard\ deviation\ of\ observation\ for\ month\ j\ at\ site\ k$$

$$i = 1, n_{jk}\ (number\ of\ observation\ for\ month\ j\ at\ site\ k)$$

$$The\ estimsted\ degrees\ of\ freedom\ N:$$

$$N = \left( \sum_k \sum_j n_{jk} \right) - (fluxes\ per\ region \times number\ of\ region \times number\ of\ month)$$

The overall reduced chi-squares for our experiments are:

|  | Mask A | Mask B | Mask C |
| --- | --- | --- | --- |
|  | YT, NT, NU | YT+NT, NU | YT+NT+NU |
| FLEXPART_EI | 1.244 | 1.237 | 1.262 |
| FLEXPART_JRA55 | 1.236 | 1.234 | 1.245 |
| WRF-STILT | 1.255 | 1.249 | 1.266 |

Given that the observations are not normally distributed (more frequent high and very high mixing ratio events than low mixing ratio events) and the limited amount of observations, there does not seem to be a strong reason to reject the model results.

We have added a paragraph on the assessment (evaluation) of our model results with reduced chi-square statistics in Section 4.6 [Comparison of modelled and observed mixing ratios (formerly Section 4.5. Comparison of prior and posterior concentrations to observations)]:

Another qualitative measure of the goodness of fit of the model to the observations is the reduced chi-square statistics (Drosg., M., 2009;. Hughes, I. G. and T. Hase, 2010). In the limit of infinite number of data points and the data are independent and normally distributed, the value of reduced chi-square should be 1. The overall reduced chi-squares for all our experiments are in a narrow range of 1.23─1.27. Given that the observations and modelled mixing ratios are not normally distributed (more frequent high and very high mixing ratio events than low mixing ratio events) and the limited amount of observations, there does not seem to be a strong reason to reject the model results

*P10, L29-30: This needs a bit more explanation, do the authors mean that they have separate variables for the biomass burning and other emissions, which are optimized simultaneously. In this case, the total number of variables would be R x 2 x number of flux time steps.*

Yes, R should be the number of all fluxes to be solved. We solved two fluxes (biomass burning and other missions) per sub-region. We have changed the sentence (note: originally on page 10, lines 14-15):

From : R is the number of sub-regions to be solved.

To : R is the number of fluxes to be solved. R is two fluxes per sub-region × number of sub-regions (i.e., 2 to 6 in this study).

*Section 3.3.2: Using only 3 regions for the optimization represents a significant aggregation error, as it is assumed that both the spatial pattern and relative magnitudes of the fluxes within each region are correct. Why was the inversion performed only for these coarse regions? Other than being different territories, are they characterized by having similar ecosystems, climate or other?*

We tried to account for the potential errors in the spatial pattern and relative magnitudes of the fluxes by using three different priors to provide a range of spatial and flux magnitude patterns. In our results, the prior flux error is smaller than the model transport error (as estimated by the different transport models used in this study).

The number of sub-regions that could be resolved by the inversion depends mainly on the amount of observations (spatial coverage density and strength of regional signals above the background variations) and magnitudes of the transport errors. In the absence of transport errors, the inversion can resolve a large number of sub-regions (an order of magnitude more in some experiments we tried). But with the present transport model errors, we begin to see unrealistic (negative fluxes) in weak flux region (Yukon) sporadically in our results. Hence, observations and model errors limit the number of sub-regions used in the inversion. The regions were defined based mainly on the geographical characteristics. Yukon has many mountains and little wetlands. Northwest Territories is mainly lowlands

with most of the wetland in the Canadian Arctic. Nunavut is a part of the Canadian Shield or Laurentian Plateau with limited wetlands.

*Section 3.3.3: Errors in the modelled background can incur errors in the posterior fluxes. Did the authors check their global modelled CH4 mixing ratios (from NIESTM) with independent observations in the northern high latitudes? Was an estimate of the uncertainty in the background made and included in the overall observation uncertainty?*

Yes, we checked the performance of NIES-TM with the independent observations. Here, we showed the comparison of model and observations for the sites, Alert, Barrow and Cold Bay, which were not used for the inversion in this study. Overall, the model captures the observed variations at synoptic scale to seasonal and long-term trend.

Yes, in our inversion, background uncertainty is implicitly included in observation error as we mentioned in the earlier response.

[Figure]

*P12, L21: The sub-region masks A to C are not defined in the text*

The sub-regions are defined in the earlier section 3.3.2. But the masks A, B and C have not been stated in the text, thought they are illustrated in Fig. S4. The sentence in Section 3.3.2 have been changed:

From:

We set up three sub-region masks for the Canadian Arctic based on three territories 1) Northwest Territories (NT), Yukon (YT), and Nunavut (NU), as shown in Fig. S4

To:

For the Canadian Arctic based on three territories, Northwest Territories (NT), Yukon (YT), and Nunavut (NU), we set up three sub-region masks, Mask A, B and C, as shown in Fig. S4.

*P12, L26: I suggest the authors state that the negative biomass burning fluxes are "spurious" since the biomass burning source cannot be negative.*

We have modified by adding a sentence as suggested:

As a result, negative mean fluxes, i.e. $CH_4$ sinks, could appear, especially in YT (Fig. 8a); the negative biomass burning fluxes are "spurious" since the biomass burning $CH_4$ source cannot be negative. However, a null-flux would be consistent within error bars.

*P15, L18-19: Do the authors mean the anomalies of the deseasonalised data? It is important to look at the anomalies in the data after the mean seasonality has been subtracted to avoid correlations with temperature between months, which would override possible correlations with temperature between years.*

Yes, the anomalies of fluxes and meteorological parameters we discussed there are the de-seasonalised data by subtracting the 4-year averaged monthly mean values. .

Figure 12: It would be interesting to see the regressions for the prior wetland emissions as well. How strongly are the prior wetland emissions correlated with the meteorological variables and how does this influence the posterior correlations?

In our study, natural $CH_4$ fluxes (wetland and other fluxes except biomass burning $CH_4$ flux) in prior emission cases, C1 and C2, are multi-year mean monthly fluxes. Therefore they have no year-to-year anomalies and no correlation with the meteorological anomalies. Only for C3, the prior wetland $CH_4$ fluxes from WetCHARTs ensemble mean exhibit inter-annual variation, the correlations with temperature and precipitation anomalies are r = 0.34 and r = 0.92 respectively.

The table below shows the correlation coefficients of the natural (wetland) posterior fluxes and the meteorological variables for individual emission scenarios along with the correlation coefficients of the prior natural fluxes. The posterior natural fluxes in C3 with WetCHARTs prior fluxes show slightly higher correlations than those in the other two cases with cyclo-stationary prior fluxes. But overall there is no significant dependency of posterior correlation on the prior wetland fluxes. This result indicates that the inter-annual variations in the posterior wetland fluxes are mainly determined by the observations, rather than by the prior fluxes. Note that as following Referee 1's suggestion, we have changed C1, C2 and C3 to VIS, GEL, and WetC respectively.

|  | Natural | | | |
| --- | --- | --- | --- | --- |
|  | temperature | | precipitation | |
|  | prior | posterior | prior | posterior |
| C1 (VIS) | 0.00 | 0.55 | 0.00 | 0.11 |
| C2 (GEL) | 0.00 | 0.54 | 0.00 | 0.07 |
| C3 (WetC) | 0.26 | 0.55 | 0.90 | 0.16 |

In the revision, we have added texts to explain the prior flux influence on the posterior flux-climate correlations in the section of "Relationship of fluxes with climate anomalies (now in Section 4.5).

In prior cases VIS and GEL, natural $CH_4$ fluxes (wetland and other fluxes except biomass burning $CH_4$ flux) are multi-year mean monthly fluxes.  Therefore these prior fluxes have no year-to-year anomalies and no correlation with the meteorological anomalies.  Only in WetC, the prior with wetland $CH_4$ fluxes from WetCHARTs ensemble mean exhibits inter-annual variations, the correlations with temperature and precipitation anomalies are $r = 0.26$ and $r = 0.90$ respectively.  The posterior natural fluxes with WetC show slightly higher correlations ($r = 0.55$ with temperature, $r=0.16$ with precipitation) than the mean correlation values. But, overall there is no clear dependency of posterior correlations on the inherent climate anomaly correlations in the prior fluxes.  This result indicates that the inter-annual variations in posterior wetland fluxes in this study are mainly determined by the observations, rather than by prior fluxes.

***Technical comments***
*P1, L26: "stronger then" should be "stronger than"*
Corrected.

*P1, L27: add "from" before "about 722 pbb"*
Added "from"

*Generally: attention should be paid to the use of articles "the" and "a" and when no article should be used at all.*
Thank you for your comment, we reviewed the text and tried to correct the usage of articles.

*P6, L14: replace "Like" with "Similar to" as "like" in this sense is very colloquial.*
Changed from "Like" to "Similar to"

*P6, L15: there are words missing in this sentence, it should be "…indicates that there is a weaker local source of CH4…" and "than around the three continental sites".*
Corrected, by adding the words as follows:

This indicates that there is a weaker local source of $CH_4$ around CBY than around the three continental sites.

*P6, L19: should be "suggested that there are on-going CH4 emissions from…"*

Changed:

From: suggested the $CH_4$ emissions from

To: suggested that there are on-going $CH_4$ emissions from

P6, L29: should be "due to the (very) short period of daylight"

Changed:

From: due to limited winter daytime

To: due to the short period of daylight

P8, L27: should be "C3 is the same as used in C2, but…"

Corrected.

P9, L31: should be "…map of climatological termite emissions"

changed

From:  a climatological emission map of termite"

To:  map of climatological termite emission"

P12, L16: change "done" to "made"

Changed.

P12, L22: should be "are shown" (not "showed")

Corrected.

*Fig. 5: should be "same as C2"*

Corrected.

P14, L25: Suggest changing the section heading to "Sensitivity tests" since there are more than one

As suggested, we changed the section heading:

From: Sensitivity test

To:   Sensitivity tests

P15, L7: should be "in winter compared to…" (not "against") and I think the authors mean "which might contribute to large uncertainties in the flux estimation"
We have changed:

From: in winter against the observed concentrations, which might have large uncertainties in flux estimation.
To: in winter compared to the observed $CH_4$, which might contribute to large uncertainties in flux estimation.

*P15, L9: change "done" to "made"*
Changed.

References:

Drosg., M., Dealing with Uncertainties: A Guide to Error Analysis , Second Edition Manfred Drosg Springer, 2009.

Gerbig, C., J. C. Lin, S. C. Wofsy, B. C. Daube, A. E. Andrews, B. B. Stephens, P. S. Bakwin, and C. A. Grainger, Toward constraining regional-scale fluxes of $CO_2$ with atmospheric observations over a continent: 1. Observed spatial variability from airborne platforms, J. Geophys. Res., 108, 4756, doi:10.1029/2002JD003018, D24, 2003.

Hughes, I. G. and T. Hase, Measurements and their Uncertainties: A Practical Guide to Modern Error Analysis Oxford, 2010.

Zhao, C., A. E. Andrews, L. Bianco, J. Eluszkiewicz, A. Hirsch, C. MacDonald, T. Nehrkorn, and M. L. Fischer (2009), Atmospheric inverse estimates of methane emissions from Central California, J. Geophys. Res., 114, D16302, doi:10.1029/2008JD011671, 2009.

---

## Author Response (AR2)

**Reply to Comments by the Co-Editor**

We thank the Co-Editor for reviewing our manuscript and pointing out the spelling errors. Beside the corrections to those spelling errors, we made other corrections and modifications. We report here our changes. The Co-Editor's comments are copied in italic and red.

*please check your manuscript again. I found many spelling errors, and I am not even a native speaker*
In a paragraph on page 2, L23-L37, we found ten misspelt words (including references). They have been corrected. Other spelling and grammar errors in the manuscript were corrected. Also, we revised the text to improve some of the more poorly worded sentences.

*check again your references and make sure they are consistent. Bergamschi is spelled Bergam_a_schi. The name Bergamschi does not appear in the references list.*
Bergamschi et al., 2013 should be Bergamaschi et al., 2013, which is in the reference list. We have corrected the short citations in the main text. Also the following six references (in Introduction) were missing in the reference list and we have added them into the list.
Schroeder et al., 2015
Miller et al., 2016; Hartery et al. 2018
Karion et al., 2016; Sasakawa et al., 2010; Chang et at., 2014

**Other changes and corrections:**

**Rephrasing:**
In order to improve the readability and clarity in our manuscript, we have revised phrases and expressions throughout the manuscript. There is no change to the scientific contents or interpretations presented previously.

[revised manuscript text omitted]